# Topological Active Inference for Task Disambiguation

Yangbo Wei [* 1 2]  Zhen Huang [* 3 2]  Shaoqiang Lu [3]  Junhong Qian [4]  Chen Wu [5]  Lei He [2]

## Abstract

In open-ended domains, natural language instructions are often *underspecified*, mapping to multiple valid yet functionally distinct latent intents. Although Large Language Models (LLMs) excel at generation, their interactive disambiguation remains limited by *semantic blindness*: they may spend clarification turns distinguishing superficial syntactic variants rather than resolving substantive intent differences. We propose *Topological Active Inference* (TAI), a geometric framework that recasts task disambiguation as *intent-manifold contraction*. TAI uses *Persistent Homology* to recover persistent intent clusters from sampled solutions, filtering short-lived syntactic variations while preserving robust semantic structure under mild separability assumptions. It then synthesizes clarifying questions as semantic separators and selects them with *Topological Expected Information Gain* (TEIG), which optimizes uncertainty reduction over intent clusters rather than individual candidates. This reduces the effective hypothesis space from $N$ sampled solutions to $K$ latent intents and yields logarithmic interaction complexity $\mathcal{O}(\log K)$ under balanced-split conditions. Experiments across code, visualization, and navigation tasks show that TAI resolves user intent with fewer turns and remains robust to noisy feedback, and smaller model scales.

## 1. Introduction

In open-ended interactions, the efficacy of LLMs is increasingly defined not by generation quality on static benchmarks,

[*]Equal contribution [1]Shanghai Jiao Tong University, Shanghai, China [2]Eastern Institute of Technology, Ningbo, China [3]University of Science and Technology of China, Hefei, China [4]Southeast University, Nanjing, China [5]Ningbo Institute of Digital Twin, Eastern Institute of Technology, Ningbo, China. Correspondence to: Chen Wu <cwu@idt.eitech.edu.cn>, Yangbo Wei <yangforever@sjtu.edu.cn>, Lei He <lhe@eitech.edu.cn>.

*Proceedings of the 43rd International Conference on Machine Learning*, Seoul, South Korea. PMLR 306, 2026. Copyright 2026 by the author(s).

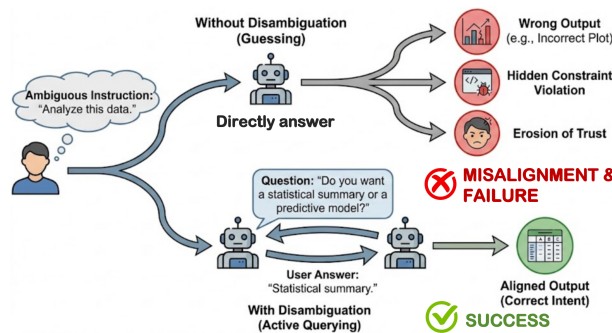

*Figure 1.* **Interactive disambiguation reduces alignment risk.** Given an underspecified instruction (e.g., Analyze this data), directly committing to one interpretation (**top**) can violate hidden constraints or yield unusable outputs. Active disambiguation (**bottom**) asks clarifying questions to shrink the hypothesis space and align the final output with the user's intent.

but by their ability to align with user intent and preferences in situ (Ouyang et al., 2022; Tamkin et al., 2023). Yet a fundamental asymmetry persists: human instructions are inherently *sparse* and *underspecified* due to the efficient, compressive nature of natural language and pragmatic inference (Clark & Brennan, 1991; Levy & Jaeger, 2007; Piantadosi et al., 2011), whereas executable solutions (e.g., code, plans) demand rigorous specificity. This mismatch is amplified because users routinely omit constraints they consider obvious (e.g., preferred toolchains, implicit safety policies, or domain assumptions), while LLM agents must still commit to concrete design choices under a space of equivalently plausible interpretations (D'Amour et al., 2022). The consequence is not merely occasional errors, but systematic *misalignment*: agents may optimize the wrong objective or violate hidden constraints, and such failures are tightly linked to specification and reward misspecification in goal-directed systems (Amodei et al., 2016; Pan et al., 2022). These failures incur real costs—extra iteration rounds, increased cognitive burden on users to correct intent, and erosion of trust—and are particularly harmful in safety-critical or high-stakes domains, where reliability, hallucinations, and broader socio-technical risks become compliance and hazard concerns (Huang et al., 2023). When an agent fills in missing context by assumption—effectively committing to one mode of a multimodal intent distribution—it risks producing outputs that are correct in form yet misaligned in function, with amplified consequences in safety-critical

settings, as shown in Fig. 1.

To mitigate this misalignment, the field has moved toward *interactive disambiguation*, where agents proactively query the user to resolve uncertainty. This paradigm is theoretically grounded in *Bayesian Experimental Design (BED)* and information-theoretic utility maximization (Lindley, 1956; Mackay, 1992), viewing the optimal question as one that maximizes the Expected Information Gain (IG) (equivalently, mutual information / expected posterior entropy reduction) over a hypothesis space (Houlsby et al., 2011; Settles, 2009). In NLP, this connects to clarification question tasks and datasets (Rao & Daumé III, 2018; Kumar & Black, 2020), and has been extended to executable generation settings such as code, where agents ask targeted clarification questions to resolve underspecified requirements (Li et al., 2023b; Mu et al., 2023; Kobalczyk et al., 2025). Yet, instantiating standard discrete BED for open-ended generation—e.g., estimating IG from a finite set of sampled *completions* and treating each distinct string as a separate hypothesis—reveals a critical pathology which we term ***semantic blindness***. Under this view, candidate solutions are discrete symbolic sequences with implicit orthogonality between distinct strings. Consequently, an agent might squander its limited interaction budget distinguishing between trivial syntactic variants (e.g., `cnt += 1` vs. `cnt = cnt + 1`) rather than resolving fundamental functional differences (e.g., `Recursive` vs. `Iterative`).

In this work, we argue that robust disambiguation requires ascending from discrete symbols to continuous geometry. We posit a *Manifold Hypothesis*: feasible solutions to an ambiguous task lie on a low-dimensional manifold in an appropriate semantic representation space, where true intents tend to form separable topological components (or low-density-separated regions) while syntactic variations induce within-component dispersion. Building on this insight, we propose *TAI*, a framework that decouples *structure discovery* from *uncertainty reduction*. TAI leverages *Persistent Homology* to recover a robust topological skeleton of the intent manifold from sampled candidates, filtering out short-lived features attributable to syntactic noise to lock onto the underlying latent intents. It then enters a closed loop, synthesizing natural language constraints that act as separating decision boundaries in representation space to efficiently bisect the probability mass over these intents.

**Contributions:** ▶ Section 2: We identify *semantic blindness* as a fundamental limitation of discrete BED for open-ended task disambiguation, and formalize task ambiguity as a *structurally fractured intent manifold* rather than an unstructured hypothesis set. ▶ Section 3: We introduce *TAI*, a geometric framework that leverages persistent homology to recover a robust topological skeleton of the solution space, separating semantic signal from syntactic noise. We

define *TEIG* to guide the selection of clarifying questions. ▶ Section A: Under mild semantic separability assumptions, maximizing TEIG induces a probabilistic mass-bisection strategy over intent clusters, yielding logarithmic query complexity $\mathcal{O}(\log K)$ —an exponential improvement over the $\mathcal{O}(N)$ complexity of sample-based discrete BED baselines.

## 2. Formalism & Background

### 2.1. The Anatomy of Ambiguity

We operate within the discrete space of natural language tokens, denoted by $\Sigma^*$. Let a problem statement $\mathcal{S} \in \Sigma^*$ serve as an instruction for an agent to generate a solution $h \in \Sigma^*$ that aligns with a ground-truth intent $\mathcal{H}^* \subset \Sigma^*$.

Unlike standard formulations that view ambiguity merely as noise, we postulate a Structural Decomposition Hypothesis. We assume that any ambiguous instruction $\mathcal{S}$ is composed of two orthogonal components $\mathcal{S} = (\mathcal{R}, \mathcal{C})$: (1) $\mathcal{R}$ denotes the Explicit Surface Constraints, representing the verifiable syntactic rules, format requirements, and literal premises stated in the text; and (2) $\mathcal{C}$ denotes the Latent Semantic Modality, representing the unobserved user intent, or world-model assumptions required to resolve the task.

**Example 1 (Open-Ended Logical Reasoning).** Consider an agent tasked with solving a logic puzzle or a "situation puzzle" (e.g., lateral thinking riddles). Here, $\mathcal{R}$ represents the *explicit scenario description* (e.g., A man walks into a bar and asks for water. The bartender points a gun at him. The man says thank you and leaves. Explain why.). The set of technically admissible solutions $\mathcal{H}$ is vast, containing thousands of logically consistent causal chains that could explain the event (e.g., the man was a spy, it was a water gun, etc.). However, the ground truth $\mathcal{H}^*$ is a single, specific narrative governed by the latent context $\mathcal{C}$ (e.g., The man had hiccups). The disambiguation process is thus a trajectory of *step-by-step convergence*: the agent must generate queries (e.g., Was the man thirsty?) to incrementally impose new constraints derived from $\mathcal{C}$, progressively pruning the branching logical trees until the solution space collapses from the broad set $\mathcal{H}$ to the unique point $\mathcal{H}^*$.

**Probabilistic Decomposition.** Let $p^*(\cdot|\mathcal{S})$ be the oracle's true likelihood function. We reformulate the generation process as a two-stage topological constraint rather than simple preference weighting:

$$p^*(h|\mathcal{S}) \propto \underbrace{\mathbb{I}_{\mathcal{R}}(h)}_{\text{Syntactic Validity}} \cdot \underbrace{\pi^*(h|\mathcal{C})}_{\text{Semantic Alignment}} \quad, \quad \forall h \in \Sigma^*. \quad (1)$$

Here, $\mathbb{I}_{\mathcal{R}}(h)$ is a hard indicator function defining the *Feasible Set* $\mathcal{H} := \{h : \mathbb{I}_{\mathcal{R}}(h) = 1\}$, which carves out the geometrically valid region in the solution space. The term

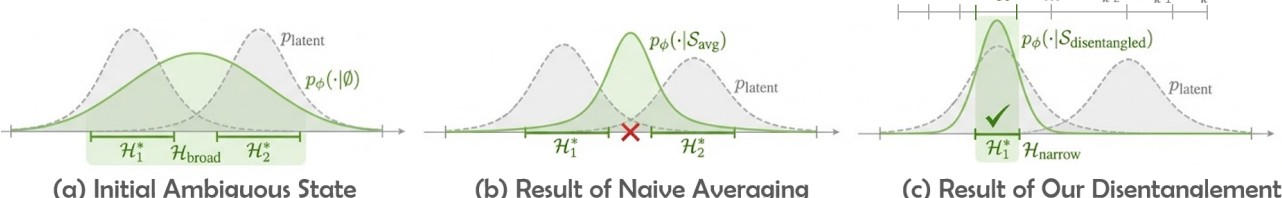

*Figure 2.* **Distributional perspectives on resolving multimodal ambiguity. (a) Initial Ambiguous State:** The latent user intent $p_{\text{latent}}$ (dashed gray) is multimodal, indicating topological ambiguity between distinct solution sets $\mathcal{H}_1^*$ and $\mathcal{H}_2^*$. **(b) Result of Naive Averaging:** Conventional methods suffer from *semantic blindness*, collapsing the model distribution $p_\phi(\cdot|\mathcal{S}_{\text{avg}})$ to a global mean. This results in a solution located in the low-validity valley (marked with $\times$) that misaligns with both true intents. **(c) Result of Our Disentanglement:** Our topological approach performs *manifold surgery*, effectively disentangling the modes. The resulting distribution $p_\phi(\cdot|\mathcal{S}_{\text{disentangled}})$ sharply aligns with a single valid intent mode $\mathcal{H}_1^*$ (marked with $\checkmark$), successfully resolving the ambiguity.

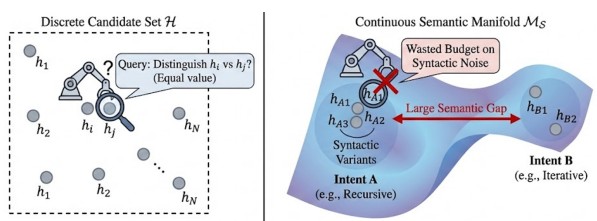

*Figure 3.* **The Limits of Discrete Disambiguation.** (a) Standard methods treat solutions as an unstructured, equidistant set. (b) In reality, solutions form semantic clusters; discrete methods suffer from 'semantic blindness' wasting queries on intra-cluster variations instead of resolving inter-cluster ambiguity.

$\pi^*(h|\mathcal{C})$ represents the *semantic density* governed by the latent context. In ambiguous tasks, $\pi^*$ is inherently multimodal; distinct realizations of $\mathcal{C}$ induce probability mass concentrations in disjoint regions of $\mathcal{H}$.

**Structural Ambiguity.** This perspective necessitates a definition of ambiguity that transcends simple underspecification. We are dealing with a fractured solution space.

**Definition 2.1** (Structural Ambiguity). Let $\mathcal{H}$ be the set of solutions satisfying explicit constraints $\mathcal{R}$. A task $\mathcal{S}$ is *structurally ambiguous* if the feasible set $\mathcal{H}$ is a proper superset of the ground truth $\mathcal{H}^*$, and $\mathcal{H}$ admits a partition into disjoint semantic clusters conditioned on latent contexts:

$$\mathcal{H} \supset \mathcal{H}^* \quad \text{and} \quad \mathcal{H} \approx \bigcup_k \mathcal{H}_k. \qquad (2)$$

This definition highlights a critical failure mode: standard agents modeling $p_\phi(\cdot|\mathcal{S})$ tend to minimize global divergence over $\mathcal{H}$. By doing so, they inherently average over the disjoint clusters $\{\mathcal{H}_k\}$, creating a distribution that covers the mean of the ambiguity rather than collapsing to the specific mode $\mathcal{H}^*$. We term this phenomenon **Mode Averaging**, which serves as the probabilistic precursor to the *Semantic Blindness* addressed in the following geometric formulation.

### 2.2. The Limits of Discrete Disambiguation

Given the structural ambiguity defined above, the objective of disambiguation is to acquire supplementary information $\mathcal{D}$ through interaction, forcing the posterior to collapse from the fractured set $\mathcal{H}$ to the ground truth $\mathcal{H}^*$.

**The Discrete Baseline.** Existing state-of-the-art methods (e.g., Kobalczyk et al., 2025) attempt to resolve this by adopting a discrete approximation strategy. They sample a finite candidate set $\mathcal{H} = \{h_i\}_{i=1}^N$ from the prior and treat it as an unstructured bag of solutions. Interaction is driven by BED, selecting a question $q$ that maximizes the standard Information Gain (IG):

$$q^* = \arg\max_q \left( \mathbb{H}(H) - \mathbb{E}_a[\mathbb{H}(H|q,a)] \right) \qquad (3)$$

**The Problem: Semantic Blindness.** This formulation suffers from a critical deficiency: by modeling $H$ as a categorical variable, it implicitly assumes that every element in $\mathcal{H}$ is *orthogonal* and *equidistant*. This implies that distinguishing between semantically identical but syntactically distinct solutions (e.g., `x = x + 1` vs. `x += 1`) yields the same information gain as distinguishing between fundamentally different logical paths (e.g., the Hiccups vs. Spy scenarios in Example 1). We term this phenomenon *Semantic Blindness* as shown in Fig. 3. Consequently, standard algorithms inherently squander their budget on eliminating high-frequency **syntactic noise** rather than resolving the low-frequency **structural ambiguity**.

**Why not direct candidate selection?** Another simple strategy is to show all sampled candidates $\mathcal{H} = \{h_i\}_{i=1}^N$ to the user and ask them to choose one. However, this shifts the burden of disambiguation to the user, who must compare many complex outputs. Such choices can also be biased by wording, order, or familiar syntax rather than true intent. TAI avoids this problem by querying the semantic distinctions between intent clusters, allowing the user to

make simple and focused judgments instead of inspecting raw candidate strings.

## 2.3. The Manifold Hypothesis

To capture the semantic affinity between solutions and overcome the blindness of discrete sets, we ascend to geometry.

**Definition 2.2** (Semantic Manifold). Let $\phi : \Sigma^* \to \mathbb{R}^d$ be a pre-trained semantic embedding mapping. For a structurally ambiguous instruction $\mathcal{S}$, the image of its feasible solution set $\mathcal{H}$ in the feature space constitutes a low-dimensional Riemannian manifold $\mathcal{M}_{\mathcal{S}} \subset \mathbb{R}^d$.

However, the existence of a manifold alone implies strictly nothing about disambiguation. To guarantee that topological features correspond to human intents rather than random artifacts, we must impose a constraint on the embedding space structure.

**Assumption 2.3** (Semantic-Syntactic Scale Separation). We assume the embedding $\phi$ preserves a multi-scale structure: the distance between distinct semantic intents (Signal) dominates the distance between syntactic variations of the same intent (Noise). Formally, for disjoint intent clusters $\mathcal{H}_i, \mathcal{H}_j \subset \mathcal{H}$, we posit:

$$\underbrace{\inf_{u \in \mathcal{H}_i, v \in \mathcal{H}_j} \|\phi(u) - \phi(v)\|}_{\delta_{\text{semantic}}} \gg \underbrace{\sup_{u,v \in \mathcal{H}_i} \|\phi(u) - \phi(v)\|}_{\epsilon_{\text{syntactic}}} \quad (4)$$

*Remark* 2.4 (Imperfect separability). Assumption 2.3 is sufficient for clean recovery, but not necessary for TAI. In practice, it is enough that inter-intent structures persist longer than intra-intent syntactic perturbations on average:

$\mathbb{E}[\ell_{\text{sem}}] > \mathbb{E}[\ell_{\text{syn}}]$. When this separation weakens, TAI may merge similar intents or over-split noisy ones, leading to coarse-grained disambiguation rather than abrupt failure.

**Geometric Implication.** This scale separation condition ($\delta_{\text{semantic}} \gg \epsilon_{\text{syntactic}}$) is critical. It implies that ambiguity manifests geometrically as a *fracture* in the manifold's connectivity. Specifically, $\mathcal{M}_{\mathcal{S}}$ is not a single connected blob, but a collection of distinct connected components separated by wide semantic voids. This structural gap provides the theoretical guarantee for using persistent homology to filter out $\epsilon$-scale noise while recovering $\delta$-scale intent structures.

## 2.4. Robust Quantification via Persistent Homology

Theoretically, the zeroth Betti number ($\beta_0$) counts the connected components of a manifold. However, computing $\beta_0$ directly on a discrete point cloud is highly unstable and dependent on the observation scale $\epsilon$: at infinitesimal scales, every point acts as an isolated component ($\beta_0 = N$), whereas at large scales, all points merge trivially ($\beta_0 = 1$).

To resolve this scale ambiguity and robustly estimate the true number of intents, we employ *Persistent Homology*.

**Intuition and Formalism.** We visualize the filtration process through a Rising Water metaphor as illustrated in Fig. 4. Treating sampled solutions as an archipelago in semantic space, we track how connectivity evolves as the observation radius $\epsilon$ increases. In this dynamic view, *noise* manifests as local syntactic variants (e.g., `count` vs. `cnt`) that merge rapidly, exhibiting short topological lifespans. Conversely, *signal* corresponds to fundamentally distinct intents (e.g., recursion vs. iteration) that remain topologically separated over a wide range of scales.

Formally, we construct a Vietoris-Rips filtration sequence $\mathcal{R}(V)$ on the embeddings $V = \phi(\mathcal{H})$ to generate the 0-th persistence diagram $\mathcal{D}_0$. Let $(b, d) \in \mathcal{D}_0$ denote a topological feature with birth time $b$ and death time $d$. We define the *robust Betti number* as the count of features whose lifespan exceeds an adaptive threshold $\tau$:

$$\beta_0^{\text{robust}} := \left| \left\{ (b, d) \in \mathcal{D}_0 \,\middle|\, d - b > \tau \right\} \right|. \quad (5)$$

This formulation effectively acts as a topological high-pass filter, discarding high-frequency oscillations (syntactic noise) while retaining the robust structural components that represent genuine semantic intents.

## 2.5. Theoretical Proof: The Semantic Redundancy Decomposition

With the introduction of $\beta_0^{\text{robust}}$, we can demonstrate why manifold-based approaches outperform discrete methods.

**Lemma 2.5** (The Semantic Redundancy Decomposition). *Let $H \in \mathcal{H}$ be a discrete solution, and let the latent variable $Z \in \{1, \ldots, \beta_0^{\text{robust}}\}$ be the index of the robust topological cluster to which the solution belongs. For any question $q$, the standard information gain $IG_{std}$ can be decomposed as:*

$$IG_{std}(q) = \underbrace{I(Z; A|q)}_{\text{Topological Gain (Signal)}} + \underbrace{I(H; A|Z, q)}_{\text{Intra-Cluster Noise}} \quad (6)$$

*Proof.* By the chain rule of mutual information, we have $I(H, Z; A|q) = I(Z; A|q) + I(H; A|Z, q)$. Since $Z$ is a deterministic function of $H$ (a solution necessarily belongs to a specific cluster), it follows that $I(H, Z; A|q) \equiv I(H; A|q)$. □

**Implication:** This lemma reveals a fundamental flaw in baseline methods: they attempt to maximize the total sum, which is often dominated by the second term (Noise). This causes the agent to become fixated on distinguishing syntactic details within clusters. In contrast, our *Topological Active Inference (TAI)* method leverages persistent homology to lock onto $Z$, directly maximizing the first term (Signal), thereby achieving efficient manifold disambiguation.

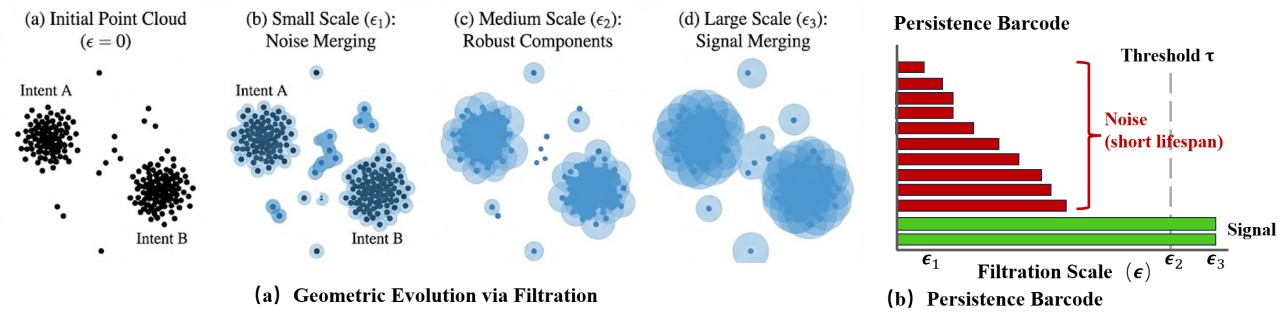

(a) Geometric Evolution via Filtration   (b) Persistence Barcode

*Figure 4.* **The mechanism of robust intent discovery via topological filtration. (Left) Geometric Evolution (The Rising Water):** We visualize the Vietoris-Rips filtration process. At small scales ($\epsilon_1$), local points representing syntactic variants (e.g., rephrased logic) rapidly merge into small connected components (noise). However, the large-scale gap between fundamentally different semantic intents (Intent A vs. Intent B) remains disconnected until a much larger scale ($\epsilon_3$). **(Right) Topological Summary (Persistence Barcode):** The geometric evolution is mapped to a barcode diagram. Each bar represents the lifespan of a connected component. High-frequency syntactic noise manifests as short, transient bars (red, lifespan $< \tau$), while true semantic intents appear as long, persistent structures (green). By filtering with an adaptive threshold $\tau$, we effectively discard the noise and recover the robust Betti number $\beta_0^{\text{robust}} = 2$.

# 3. Method: Topological Active Inference

Building upon the Manifold Hypothesis established in Section 2, we propose *Topological Active Inference (TAI)*.

TAI resolves an ambiguous instruction by organizing sampled solutions into semantic intent clusters, asking clarifying questions that separate competing clusters, and updating the belief over user intent after each response. This turns open-ended clarification into a structured loop of intent discovery, question selection, and posterior refinement.

## 3.1. Recovering Intent Clusters

Since the true intent manifold $\mathcal{M}$ is implicit, our first objective is to reconstruct its topological skeleton.

**Embedding & Point Cloud Construction.** Let $\phi : \Sigma^* \rightarrow \mathbb{R}^d$ be a pre-trained encoder (e.g., OpenAI `text-embedding-3`). We sample $N$ candidate solutions $\mathcal{H} = \{h_i\}_{i=1}^N$ from the LLM's initial posterior to construct a semantic point cloud $V = \{\phi(h_i)\}_{i=1}^N$.

**Topological Denoising Guarantee.** We construct a Vietoris–Rips filtration $\mathcal{R}(V)$ and compute the 0-th persistence barcode. To justify filtering short-lived features, we assume bounded geometric distortion in the embedding.

**Assumption 3.1** (Effective $(\epsilon, \delta)$-Separability)**.** Let $\mathcal{H}^* = \{I_1, \ldots, I_K\}$ denote the latent intents. After density-based filtering, let $V$ denote the remaining point set and let $I_k \subseteq V$ be the points of intent $k$. We assume the embedding $\phi$ yields separated sets such that $\text{diam}(\phi(I_k)) \leq \epsilon_{\text{noise}}$ for all $k$ (intra-intent compactness), and $\text{dist}(\phi(I_k), \phi(I_\ell)) \geq \delta_{\text{signal}}$ for all $k \neq \ell$ (inter-intent separability).

**Proposition 3.2** (Topological Recovery)**.** *If the filtered embedding satisfies Assumption 3.1 with $\delta_{signal} > \epsilon_{noise}$, then there exists a persistence threshold $\tau \in (\epsilon_{noise}, \delta_{signal})$ such* *that the robust Betti number $\beta_0^{robust}$ consistently recovers the true number of latent intents, i.e., $\beta_0^{robust} = |\mathcal{H}^*|$.*

*Proof.* By the definition of the Vietoris–Rips complex $\mathcal{R}_\alpha(V)$, edges are added between points $u, v$ if $\|\phi(u) - \phi(v)\| \leq \alpha$. The intra-intent compactness assumption implies that for any scale $\alpha \geq \epsilon_{\text{noise}}$, all points within the effective support of a cluster $I_k$ form a single connected component (Edelsbrunner & Harer, 2010); thus, all intra-cluster topological features in the persistence diagram $\mathcal{D}_0$ have death times $d \leq \epsilon_{\text{noise}}$. Conversely, the inter-intent separability guarantees that distinct clusters remain topologically disjoint for all $\alpha < \delta_{\text{signal}}$. Following the Stability Theorem for persistence diagrams (Cohen-Steiner et al., 2005), this spectral gap ensures the persistence intervals partition strictly into noise sets with lifetimes $\ell \leq \epsilon_{\text{noise}}$ and signal sets with lifetimes $\ell \geq \delta_{\text{signal}}$. Consequently, for any threshold $\tau \in (\epsilon_{\text{noise}}, \delta_{\text{signal}})$, the robust count satisfies $\beta_0^{\text{robust}} = |\{(0, d) \in \mathcal{D}_0 \mid d > \tau\}| = K$, effectively recovering the cardinality of $\mathcal{H}^*$. $\square$

Based on this proposition, we compute $\mathcal{R}(V)$ once and extract $K$ *Robust Semantic Clusters* $\mathcal{C} = \{C_1, \ldots, C_K\}$. These clusters serve as the fixed geometric anchors for the subsequent active inference loop.

## 3.2. Topology-Guided Question Generation

Having identified the clusters $\mathcal{C}$, TAI generates questions that separate competing intent clusters rather than searching blindly. At each round, TAI selects the most competitive clusters under the current belief and uses contrastive prompting to synthesize candidate questions.

For the default binary setting, let $A_q \in \{0, 1\}$ denote the answer to question $q$. A good question should give different

answers for two clusters $C_a$ and $C_b$:

$$P(A_q = 1 \mid h \in C_a) \geq 1 - \eta, \quad P(A_q = 1 \mid h \in C_b) \leq \eta, \tag{7}$$

where $\eta \in [0, 0.5)$ allows for uncertainty in the LLM's prediction. Thus, each question acts as a soft separator between intent clusters. If all cluster pairs are considered, the candidate pool has size $\mathcal{O}(K^2)$. In practice, we generate questions only for the top competing clusters. Although we mainly use binary questions for simplicity, the same idea can be extended to multi-choice questions; Section 3.3 defines TEIG over a general answer space $\mathcal{A}_q$.

### 3.3. Scoring Questions with TEIG

To select the optimal question $q^* \in \mathcal{Q}_{\text{topo}}$, we maximize the mutual information between the user's latent intent variable $Z \in \{1, \ldots, K\}$ and the answer induced by question $q$.

We define the belief state $P_t(z)$ as the probability that the user's true intent corresponds to cluster $C_z$. Let $\mathcal{A}_q$ denote the answer space of question $q$, which can be binary, multi-choice, or include an uncertain response. The *Topological Expected Information Gain (TEIG)* is:

$$
\begin{aligned}
\text{TEIG}(q) = \sum_{z=1}^{K} P_t(z) \sum_{a \in \mathcal{A}_q} P(a \mid z, q) \\
\cdot \log \frac{P(a \mid z, q)}{\sum_{z'=1}^{K} P(a \mid z', q) \, P_t(z')}.
\end{aligned}
\tag{8}
$$

In our main implementation, we use $\mathcal{A}_q = \{0, 1\}$ because binary semantic judgments are simple and reliable for users. The same formulation also supports richer feedback, such as multi-choice questions or ternary responses $\mathcal{A}_q = \{\text{Yes}, \text{No}, \perp\}$, where $\perp$ means "I don't know".

**Connection to Optimal Search.** As proven in Theorem A.1 and Theorem A.3, maximizing TEIG minimizes the expected posterior uncertainty over intent clusters. For binary questions, this objective favors near-balanced splits of the current belief mass, i.e., $P(A_q = 1) \approx 0.5$. Thus, each informative question removes a constant fraction of the remaining ambiguity, yielding a logarithmic convergence rate of $\mathcal{O}(\log K)$ under the separability conditions.

### 3.4. Updating Beliefs from User Feedback

After the intent clusters $\mathcal{C}$ are recovered, TAI keeps the cluster geometry fixed and updates only the belief over clusters during interaction. This separates the static structure of the solution space from the dynamic uncertainty about the user's true intent.

**Posterior update.** At round $t$, after asking question $q_t$ and observing the user's response $\tilde{a}_t \in \mathcal{A}_{q_t}$, TAI updates the belief over intent clusters using Bayes' rule:

$$P_{t+1}(z) = \frac{P(\tilde{a}_t \mid z, q_t) P_t(z)}{\sum_{z'=1}^{K} P(\tilde{a}_t \mid z', q_t) P_t(z')}. \tag{9}$$

Here, $P(\tilde{a}_t \mid z, q_t)$ measures how likely cluster $C_z$ is to produce the observed answer. Since each question is designed to separate competing clusters, the likelihood is high for clusters consistent with the response and low for inconsistent ones. Thus, the update concentrates probability mass on the cluster that best matches the user's intent.

This update is soft rather than eliminative: no cluster must be permanently discarded after a single response. This is important under noisy or uncertain feedback. For example, if the user answers "I don't know", we can model it as an uncertain response $\tilde{a}_t = \perp$ with cluster-independent likelihood, which leaves the posterior nearly unchanged and simply triggers another question.

**Stopping condition.** The interaction stops when one cluster dominates the posterior:

$$P_t(k^*) > 1 - \rho, \tag{10}$$

where $C_{k^*}$ is the most likely intent cluster and $\rho$ is a confidence tolerance. We distinguish question selection from termination: TEIG favors balanced splits to reduce uncertainty quickly, but TAI stops only when the posterior becomes sufficiently concentrated, rather than when a cluster merely obtains a simple majority.

**Complexity analysis.** TAI separates one-time structure discovery from online interaction. It first samples $N$ candidates, embeds them, and computes persistent homology once to recover $K$ robust intent clusters. During interaction, the topology is fixed; TAI only generates questions for the most competitive clusters and updates a $K$-dimensional belief vector. If all cluster pairs are considered, question generation costs $\mathcal{O}(K^2)$, with $K \ll N$ in practice. Under probability-bisection selection, the number of interaction turns scales as $\mathcal{O}(\log K)$, in contrast to discrete baselines that reason over $N$ individual candidates.

## 4. Experiments

We evaluate TAI along five research questions covering efficiency, mechanism, semantic sensitivity, model generalization, and reliability under imperfect feedback.

**Research Questions.** ▶ **RQ1 (Efficiency):** Can TAI resolve user intent with fewer interaction turns than entropy-based baselines? ▶ **RQ2 (Mechanism):** Does the gain come from topological skeletonization rather than simple clustering or larger sampling? ▶ **RQ3 (Sensitivity):** Can TAI distinguish true functional differences from superficial

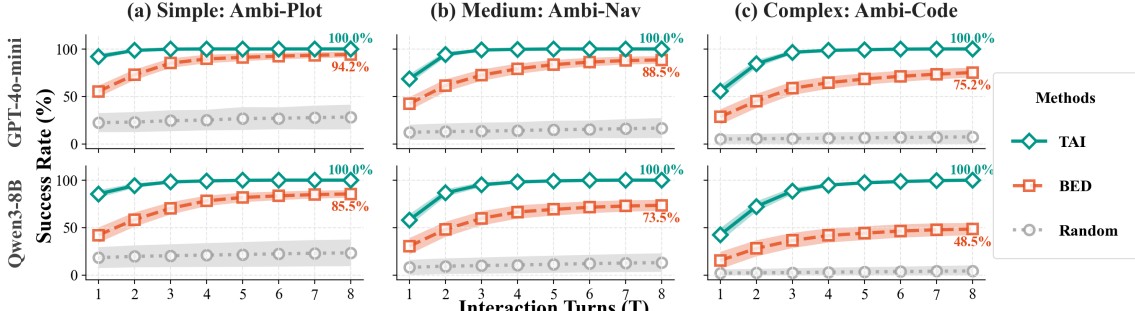

*Figure 5.* **Efficiency comparison across Ambi-Bench domains.** TAI reaches high success within a few turns across all domains. In harder cases, Standard BED plateaus due to semantic blindness, while Random-Ask fails to gather useful intent information.

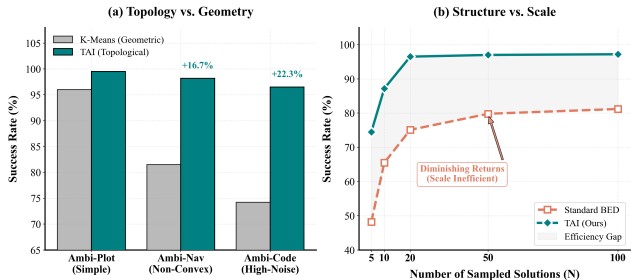

*Figure 6.* **Quantitative ablation studies. (a) Topology vs. Geometry:** TAI outperforms K-Means significantly on non-convex tasks (Ambi-Nav) and high-noise tasks (Ambi-Code). **(b) Structure vs. Scale:** Simply increasing sample size ($N$) for Standard BED yields diminishing returns (red dashed line), whereas TAI maintains high accuracy even with sparse samples ($N = 20$).

syntactic variations? ▶ **RQ4 (Generalization):** Does TAI remain effective on smaller open-weight models? ▶ **RQ5 (Imperfect Feedback):** Is TAI reliable when user feedback is noisy, uncertain, or partially incorrect?

## 4.1. Experimental Setup

**Benchmark.** We introduce *Ambi-Bench*, a curated benchmark for task disambiguation with three domains. *Ambi-Code* is derived from HumanEval by removing key functional constraints, such as time complexity or library requirements, so that each prompt admits multiple valid solution clusters. *Ambi-Plot* contains underspecified visualization tasks where the hidden intent concerns chart type or styling choices, such as heatmaps versus scatter plots. *Ambi-Nav* is a GridWorld planning domain where instructions such as "reach the goal" admit different paths under latent objectives such as safety or speed.

**User Feedback.** In the main experiments, we use an LLM-based oracle with access to the ground-truth constraint $\mathcal{C}^*$, following prior work. The oracle provides binary answers for closed-ended questions and concise specifications for open-ended queries. To test more realistic interaction, RQ5 further relaxes this setting by introducing answer flipping,

"I don't know" responses, and mixed feedback noise.

**Baselines.** We compare TAI with representative disambiguation strategies, including *Random-Ask*, which samples clarification questions uniformly at random, and *Standard BED*, which selects questions by maximizing expected information gain over sampled candidates. For stronger comparisons, we also evaluate embedding-aware and clustering-enhanced BED variants in the ablation analyses.

**Implementation Details.** We use a pretrained embedding model for semantic mapping and `GUDHI` to compute Vietoris–Rips persistence barcodes. Unless otherwise specified, we set the sample size to $N = 20$, the question candidate pool to $M = 5$, and the interaction budget to $T = 10$. Full configurations are provided in Appendix F.2.

## 4.2. Results and Analysis

### 4.2.1. RQ1: EFFICIENCY & EFFECTIVENESS

We begin our analysis by addressing the core performance metric: can the agent align with user intent both *rapidly* and *accurately*? The experimental results in Figure 5 reveal a distinct performance disparity between TAI and baselines.

**The Agility of Convergence.** TAI demonstrates exceptional efficiency, requiring only 1 to 3 turns to achieve $> 95\%$ success across all domains. This empirical evidence validates our theoretical reduction of query complexity from linear $\mathcal{O}(N)$ to logarithmic $\mathcal{O}(\log K)$. In stark contrast, the *Random-Ask* baseline stagnates as a **flatline** (Figure 5) even after extended interactions ($T = 8$), confirming that unstructured probing yields negligible information gain in high-dimensional spaces.

**The Ceiling of Alignment.** Beyond speed, TAI establishes a superior precision benchmark. In complex scenarios (Figure 5(c)), Standard BED hits a hard **performance ceiling** of $\sim$**75%**, constrained by *semantic blindness*—it squanders queries distinguishing syntactic noise rather than

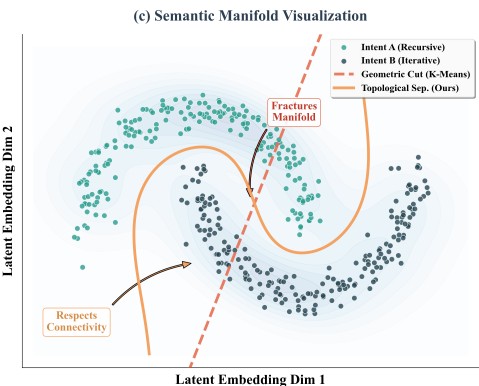

Figure 7. **Visualizing Semantic Blindness.** A t-SNE projection of the solution space illustrates why geometric clustering fails.

functional intent. Conversely, TAI leverages its topological skeleton to filter such irrelevancies, maintaining a robust trajectory toward **100%** convergence. This confirms that structural insight, rather than a larger interaction budget, is the decisive factor for resolving ambiguity.

### 4.2.2. RQ2: MECHANISM (ABLATION STUDY)

Having established TAI's efficiency, we investigate the causal mechanism behind its success. We conduct two ablation studies to determine whether the performance gain stems from *topological* insight or merely from geometric clustering and increased sampling scale.

**The Necessity of Topology (vs. Geometric Rigidity).** We first replace persistent homology with K-Means to test if simple geometric clustering suffices. Results indicate a significant performance drop (Fig. 6(a)). This disparity highlights the non-convex nature of intent manifolds: unlike K-Means, which assumes spherical clusters in Euclidean space, latent user intents often form elongated or irregular structures. As visualized in Fig. 7, K-Means erroneously fractures these continuous manifolds into fragmented clusters, triggering unnecessary queries. In contrast, TAI relies on *connectivity* rather than distance, allowing it to adapt fluidly to the manifold's intrinsic shape and correctly group syntactically diverse but topologically connected solutions.

**Beyond Brute-Force Sampling (Structure > Scale).** Second, we examine whether simply increasing the sample size ($5 \times N$) can replicate TAI's performance via "brute-force" enumeration. Contrary to intuition, massive sampling yields rapidly diminishing returns for the Standard BED baseline (Fig. 6(b)). Without structural guidance, additional samples merely introduce more *syntactic variants*, effectively drowning the agent in high-frequency noise. This confirms that TAI's advantage derives not from computational scaling, but from its function as a **high-pass filter**.

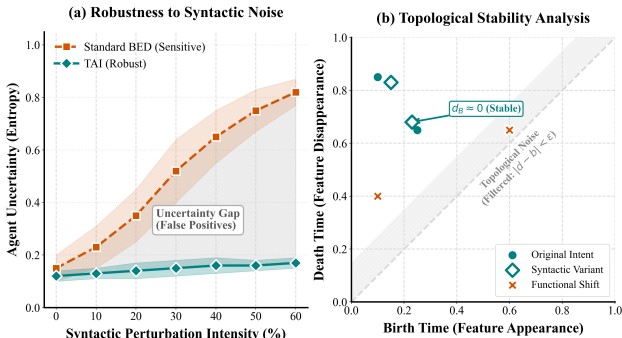

Figure 8. **Semantic robustness and sensitivity.** **(a)** TAI is robust to syntactic noise, while the baseline shows rising entropy. **(b)** Persistence-space analysis explains why: syntactic perturbations induce negligible topological change ($d_B \approx 0$), whereas true functional shifts cause large deviations ($d_B \gg \epsilon$).

Table 1. **Robustness to syntactic attacks.** We measure the entropy increase ($\Delta$Entropy) and success-rate drop ($\Delta$SR) under different perturbations. TAI remains stable.

| Perturbation Method | Standard BED (Baseline) | | TAI (Ours) | |
|---|---|---|---|---|
| | $\Delta$ **Entropy** ↑ | $\Delta$ **SR** ↓ | $\Delta$ **Entropy** ↑ | $\Delta$ **SR** ↓ |
| Variable Renaming | +0.42 | -15.4% | +0.03 | -0.5% |
| Dead Code Insertion | +0.38 | -12.1% | +0.05 | -0.8% |
| Format Reordering | +0.25 | -8.6% | +0.01 | -0.2% |
| Hybrid Attack (All) | +0.65 | -28.5% | +0.08 | -1.2% |

### 4.2.3. RQ3: SEMANTIC SENSITIVITY & ROBUSTNESS

If efficiency represents the speed of an agent, semantic sensitivity defines its discernment. Real-world interactions are rife with syntactic variations that convey identical intents, yet subtle logical shifts can fundamentally alter function. An ideal disambiguation framework requires a dual nature: insensitivity to *Syntactic Perturbations* (Invariance) but acute sensitivity to *Functional Differences*.

**Resisting Syntactic Camouflage.** We injected randomized noise into the Ambi-Code dataset (e.g., variable renaming, dead code insertion) to generate syntactically diverse but semantically identical variants. As illustrated in Figure 8a, Standard BED suffers from severe **false positive** entropy spikes as perturbation intensity increases. Deceived by superficial distances in the embedding space, the baseline misinterprets syntactic noise as divergent intents, squandering precious queries on trivialities.

**Topological Invariance as Semantic Fingerprint.** In contrast, TAI demonstrates intrinsic topological stability. As visualized in the persistence diagram in Figure 8b, despite drastic textual perturbations, the topological features of the variant map closely to the original intent, resulting in a negligible *Bottleneck Distance* ($d_B \approx 0$). This confirms that TAI captures the *topological invariants* of the intent manifold—serving as a stable **semantic fingerprint** that persists across surface-level distortions, ensuring the agent focuses

*Table 2.* **Robustness to noisy oracle feedback.** Success Rate (%) under three noise models. TAI degrades gracefully; BED collapses. Parenthetical green/red values = $\Delta$SR from the clean baseline. Combined level = $p_{flip}$ / $p_{unk}$.

| Noise Model | Level | Ambi-Code | | Ambi-Plot | | Ambi-Nav | |
|---|---|---|---|---|---|---|---|
| | | TAI | BED | TAI | BED | TAI | BED |
| *Clean* ($p$=0) | | **100** | 75.0 | **100** | 94.2 | **100** | 88.5 |
| Answer Flipping | $p_{flip}$=.10 | **96.0** (−4) | 61.0 (−14) | **98.3** (−2) | 84.5 (−10) | **97.1** (−3) | 76.2 (−12) |
| | $p_{flip}$=.20 | **91.0** (−9) | 48.0 (−27) | **95.1** (−5) | 72.3 (−22) | **92.8** (−7) | 61.5 (−27) |
| | $p_{flip}$=.30 | **79.0** (−21) | 34.0 (−41) | **89.5** (−11) | 57.8 (−36) | **83.6** (−16) | 46.2 (−42) |
| "I Don't Know" | $p_{unk}$=.20 | **96.0** (−4) | 63.5 (−12) | **98.7** (−1) | 87.1 (−7) | **97.5** (−3) | 78.6 (−10) |
| | $p_{unk}$=.40 | **90.2** (−10) | 46.8 (−28) | **94.0** (−6) | 74.5 (−20) | **91.8** (−8) | 63.2 (−25) |
| Combined | .10 / .10 | **94.5** (−6) | 57.2 (−18) | **97.6** (−2) | 82.3 (−12) | **96.0** (−4) | 72.8 (−16) |
| | .20 / .20 | **85.4** (−15) | 38.6 (−36) | **91.2** (−9) | 63.7 (−31) | **87.5** (−13) | 51.4 (−37) |
| **Avg. Robustness Gap** | | **+38.8** | | **+21.6** | | **+30.3** | |

*Table 3.* **Model generalization (Qwen 0.6B–14B).** Sparklines visualize convergence trajectories over interaction turns ($T = 1 \rightarrow$ 5). While Standard BED degrades into a flatline on smaller models due to limited reasoning, TAI maintains a robust ascent even on 0.6B, effectively acting as a cognitive scaffold.

| Model Scale | Standard BED | | TAI (Ours) | | Robustness |
|---|---|---|---|---|---|
| | SR (%) | Trajectory | SR (%) | Trajectory | Gap ($\Delta$) |
| Qwen-14B | 78.5 | | 99.2 | | +20.7% |
| Qwen-8B | 62.1 | | 98.5 | | +36.4% |
| Qwen-4B | 45.3 | | 95.8 | | +50.5% |
| Qwen-1.7B | 28.4 | | 81.2 | | +52.8% |
| Qwen-0.6B | 15.2 | | 64.5 | | +49.3% |

solely on logical structure rather than syntactic form.

### 4.2.4. RQ4: MODEL GENERALIZATION

Finally, we address the universality of our framework: is TAI merely a luxury tool contingent upon the emergent capabilities of large-scale models, or a generalizable framework accessible to smaller models? To verify this, we evaluate performance across a spectrum of model capabilities.

**Cognitive Scaffolding.** As detailed in Table 3 , results reveal a stark contrast. Standard BED suffers a **catastrophic collapse** on smaller models (dropping to ∼ 15% SR on 0.6B), as entropy-based methods fail without precise posterior estimation. Conversely, TAI demonstrates remarkable **Model Agnosticism**, maintaining high alignment success (> 64%) even on the smallest architecture. TAI effectively functions as a **cognitive scaffold** utilizing topological structure to bridge the reasoning gap. This confirms that TAI's efficacy stems from geometric reduction of the search space rather than raw computational power.

### 4.2.5. RQ5: RELIABILITY UNDER FEEDBACK

Unlike RQ3, which studies robustness to syntactic perturbations in candidate solutions, this experiment evaluates

whether TAI remains reliable when user feedback is imperfect. As shown in Table 2, we consider answer flipping, "I don't know" responses, and their combination. Across all three domains, TAI degrades much more slowly than BED. Under the strongest combined noise ($p_{flip}$ = .20, $p_{unk}$ = .20), TAI still achieves 85.4%, 91.2%, and 87.5% success rates on Ambi-Code, Ambi-Plot, and Ambi-Nav, respectively, while BED drops to 38.6%, 63.7%, and 51.4%.

This reliability comes from TAI's cluster-level belief update. A noisy or uncertain answer only softly shifts probability mass between semantic clusters, rather than permanently eliminating individual candidates. In contrast, BED reasons over discrete candidates, so early feedback errors can distort the posterior and misguide later questions.

## 5. Conclusion

We introduced *TAI*, a topological framework for resolving ambiguous user instructions. By recovering robust intent clusters from sampled solutions and selecting clarifying questions with Topological Expected Information Gain, TAI shifts disambiguation from discrete candidate elimination to structured intent-space search. This yields efficient $\mathcal{O}(\log K)$ interaction, improves robustness to syntactic and feedback noise, and provides a useful scaffold across scales.

TAI depends on candidate coverage and embedding quality: missing intent modes cannot be recovered, and weak semantic separation can lead to coarse or noisy clusters. Our evaluation remains small-scale. Future work should make the topology adaptive through resampling and online refinement, and combine TAI with free-form dialogue so that agents can switch between natural conversation and topology-guided clarification. More broadly, we believe topological intent modeling offers a path toward agents that actively uncover the hidden structure of user goals rather than merely responding to underspecified prompts.

## Impact Statement

This paper proposes an interactive task disambiguation method for improving the reliability of LLM assistants under underspecified user instructions. By focusing clarification on semantically meaningful intent differences rather than superficial syntactic variants, the method can reduce unnecessary interaction turns, lower user burden, and help avoid downstream failures caused by committing to an incorrect interpretation.

The approach may benefit applications such as code generation, data analysis, and planning, where misunderstandings can be costly. However, more efficient disambiguation may also amplify misuse or over-reliance if deployed without proper safeguards. We therefore view TAI as a component of broader human-in-the-loop and safety-aware systems. The method does not require additional user data beyond standard interaction signals and does not explicitly model sensitive personal attributes.

## Acknowledgements

This work by Chen Wu was supported by the Science and Technology Innovation Key R&D Program of Chongqing (CSTB2025TIAD-STX0016) and the Ningbo Yongjiang Talent Programme (2025A-169-G).

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

## A. Theoretical Analysis

To substantiate the algorithmic advantages of Topological Active Inference (TAI), we provide rigorous theoretical bounds regarding query complexity and noise robustness.

**Theorem A.1** (Logarithmic Query Complexity under $\gamma$-Effective Separability). *Let $\mathcal{C} = \{C_1, \ldots, C_K\}$ be the set of $K$ latent semantic intent clusters, and let $\mu$ be a fixed prior measure over $\mathcal{C}$. For a version space $\mathcal{V}_t \subseteq \mathcal{C}$, define its mass as*

$$\mu(\mathcal{V}_t) = \sum_{C \in \mathcal{V}_t} \mu(C).$$

*Assume the $\gamma$-**effective separability condition**: for any non-singleton version space $\mathcal{V}_t$, there exists a question $q \in \mathcal{Q}$ that partitions $\mathcal{V}_t$ into two answer-consistent subsets $\mathcal{V}_t^{yes}$ and $\mathcal{V}_t^{no}$ such that*

$$\min \{\mu(\mathcal{V}_t^{yes}), \mu(\mathcal{V}_t^{no})\} \geq \gamma \, \mu(\mathcal{V}_t), \qquad \gamma \in (0, 0.5]. \tag{11}$$

*If TAI selects such a split at each step, then the number of queries required to reduce the remaining version-space mass below a target resolution $\epsilon$ is bounded by*

$$T_\epsilon \leq \left\lceil \frac{\ln(1/\epsilon)}{-\ln(1 - \gamma)} \right\rceil \leq \left\lceil \frac{1}{\gamma} \ln \frac{1}{\epsilon} \right\rceil. \tag{12}$$

*In particular, under a uniform prior $\mu(C_i) = 1/K$, setting $\epsilon = 1/K$ isolates a single intent cluster, giving*

$$T = \mathcal{O}\left(\frac{1}{\gamma} \log K\right). \tag{13}$$

*Proof.* We view TAI as a generalized binary search procedure over semantic intent clusters. Initially, all clusters are feasible, so $\mathcal{V}_0 = \mathcal{C}$ and $\mu(\mathcal{V}_0) = 1$.

At step $t$, the selected question partitions the current version space into two subsets, $\mathcal{V}_t^{yes}$ and $\mathcal{V}_t^{no}$. After observing the oracle answer $a_t \in \{yes, no\}$, the next version space is the answer-consistent subset:

$$\mathcal{V}_{t+1} = \mathcal{V}_t^{a_t}.$$

In the worst case, the true intent lies in the larger side of the partition. Therefore,

$$
\begin{aligned}
\mu(\mathcal{V}_{t+1}) &= \mu(\mathcal{V}_t^{a_t}) \\
&\leq \max \{\mu(\mathcal{V}_t^{yes}), \mu(\mathcal{V}_t^{no})\} \\
&= \mu(\mathcal{V}_t) - \min \{\mu(\mathcal{V}_t^{yes}), \mu(\mathcal{V}_t^{no})\} \\
&\leq \mu(\mathcal{V}_t) - \gamma \, \mu(\mathcal{V}_t) \\
&= (1 - \gamma)\mu(\mathcal{V}_t).
\end{aligned}
\tag{14}
$$

Applying this recurrence for $T$ steps gives

$$
\begin{aligned}
\mu(\mathcal{V}_T) &\leq (1 - \gamma)\mu(\mathcal{V}_{T-1}) \\
&\leq (1 - \gamma)^2 \mu(\mathcal{V}_{T-2}) \\
&\leq \cdots \\
&\leq (1 - \gamma)^T \mu(\mathcal{V}_0) \\
&= (1 - \gamma)^T.
\end{aligned}
\tag{15}
$$

To reduce the version-space mass below a target resolution $\epsilon$, it is sufficient to require

$$
\begin{aligned}
(1 - \gamma)^T &\leq \epsilon \\
\Longleftrightarrow \quad T \ln(1 - \gamma) &\leq \ln \epsilon \\
\Longleftrightarrow \quad T &\geq \frac{\ln(1/\epsilon)}{-\ln(1 - \gamma)},
\end{aligned}
\tag{16}
$$

where the inequality reverses after division because $\ln(1 - \gamma) < 0$. Thus, choosing

$$T_\epsilon = \left\lceil \frac{\ln(1/\epsilon)}{-\ln(1 - \gamma)} \right\rceil \tag{17}$$

is sufficient to ensure $\mu(\mathcal{V}_T) \leq \epsilon$.

Finally, since $-\ln(1 - \gamma) \geq \gamma$ for $\gamma \in (0, 1)$, we have

$$\begin{aligned} T_\epsilon &= \left\lceil \frac{\ln(1/\epsilon)}{-\ln(1 - \gamma)} \right\rceil \\ &\leq \left\lceil \frac{1}{\gamma} \ln \frac{1}{\epsilon} \right\rceil. \end{aligned} \tag{18}$$

Under a uniform prior, each cluster has mass $1/K$. Therefore, reaching $\mu(\mathcal{V}_T) \leq 1/K$ implies that the remaining version space contains at most one cluster. Substituting $\epsilon = 1/K$ yields

$$T \leq \left\lceil \frac{\ln K}{-\ln(1 - \gamma)} \right\rceil \leq \left\lceil \frac{1}{\gamma} \ln K \right\rceil = \mathcal{O}\left( \frac{1}{\gamma} \log K \right). \tag{19}$$

This establishes logarithmic query complexity in the number of latent intent clusters. $\qquad\square$

**Theorem A.2** (Noise Immunity of TEIG). *Let $q_{\text{noise}}$ be a purely syntactic question whose answer distribution is independent of the latent intent cluster $Z$. That is, for all $z \in \{1, \ldots, K\}$ and all $a \in \mathcal{A}$,*

$$P(a \mid Z = z, q_{\text{noise}}) = \rho(a), \tag{20}$$

*where $\rho$ is a fixed answer distribution. Then the Topological Expected Information Gain of $q_{\text{noise}}$ is zero:*

$$\text{TEIG}_t(q_{\text{noise}}) = 0. \tag{21}$$

*Proof.* By definition, TEIG is the mutual information between the latent intent $Z$ and the answer $A$ conditioned on the question:

$$\begin{aligned} \text{TEIG}_t(q_{\text{noise}}) &= I_t(Z; A \mid q_{\text{noise}}) \\ &= \sum_{z=1}^{K} P_t(z) \sum_{a \in \mathcal{A}} P(a \mid z, q_{\text{noise}}) \log \frac{P(a \mid z, q_{\text{noise}})}{P(a \mid q_{\text{noise}})}. \end{aligned} \tag{22}$$

Since $q_{\text{noise}}$ is independent of $Z$, we have

$$\begin{aligned} P(a \mid q_{\text{noise}}) &= \sum_{z'=1}^{K} P_t(z') P(a \mid z', q_{\text{noise}}) \\ &= \sum_{z'=1}^{K} P_t(z') \rho(a) \\ &= \rho(a) \sum_{z'=1}^{K} P_t(z') \\ &= \rho(a). \end{aligned} \tag{23}$$

Substituting this marginal distribution back into the TEIG expression gives

$$\begin{aligned} \text{TEIG}_t(q_{\text{noise}}) &= \sum_{z=1}^{K} P_t(z) \sum_{a \in \mathcal{A}} \rho(a) \log \frac{\rho(a)}{\rho(a)} \\ &= \sum_{z=1}^{K} P_t(z) \sum_{a \in \mathcal{A}} \rho(a) \log 1 \\ &= 0. \end{aligned} \tag{24}$$

Here, terms with $\rho(a) = 0$ are interpreted under the standard convention $0 \log 0 = 0$. Therefore, a question that only probes syntactic variation has zero TEIG because it induces no distributional difference across topological intent clusters. $\qquad\square$

**Theorem A.3** (Optimality of Balanced Partition). *At interaction step $t$, assume that a question $q \in \mathcal{Q}$ induces an ideal deterministic binary partition over the semantic intent clusters. Specifically, there exists a cluster-level answer map*

$$g_q : \{1, \ldots, K\} \to \{0, 1\}, \tag{25}$$

*such that all candidates within the same cluster $C_z$ yield the same answer $A = g_q(z)$. Let*

$$\mathcal{Z}_1(q) = \{z : g_q(z) = 1\}, \qquad \mathcal{Z}_0(q) = \{z : g_q(z) = 0\}. \tag{26}$$

*Then maximizing TEIG is equivalent to maximizing the binary entropy of the induced answer distribution. In particular, if an exact balanced split is feasible, the optimal question satisfies*

$$\sum_{z \in \mathcal{Z}_1(q^*)} P_t(z) = \frac{1}{2}. \tag{27}$$

*If no exact balanced split exists, the optimal question is the feasible split whose positive-side probability mass is closest to $\frac{1}{2}$.*

*Proof.* Recall that TEIG is the mutual information between the latent intent variable $Z$ and the answer $A$:

$$\begin{aligned}
\text{TEIG}_t(q) &= I_t(Z; A \mid q) \\
&= \mathbb{H}_t(A \mid q) - \mathbb{H}_t(A \mid Z, q).
\end{aligned} \tag{28}$$

Under the ideal deterministic partition assumption, the answer is fixed once the intent cluster is known:

$$P(A = a \mid Z = z, q) = \mathbb{1}\{a = g_q(z)\}. \tag{29}$$

Therefore, the conditional entropy vanishes:

$$\begin{aligned}
\mathbb{H}_t(A \mid Z, q) &= -\sum_{z=1}^{K} P_t(z) \sum_{a \in \{0,1\}} P(a \mid z, q) \log P(a \mid z, q) \\
&= -\sum_{z=1}^{K} P_t(z) \sum_{a \in \{0,1\}} \mathbb{1}\{a = g_q(z)\} \log \mathbb{1}\{a = g_q(z)\} \\
&= 0.
\end{aligned} \tag{30}$$

Thus, maximizing TEIG reduces to maximizing the marginal entropy of the answer:

$$\text{TEIG}_t(q) = \mathbb{H}_t(A \mid q). \tag{31}$$

Let

$$p_q = P_t(A = 1 \mid q). \tag{32}$$

By the law of total probability and the deterministic partition assumption,

$$\begin{aligned}
p_q &= \sum_{z=1}^{K} P_t(z) P(A = 1 \mid Z = z, q) \\
&= \sum_{z=1}^{K} P_t(z) \mathbb{1}\{g_q(z) = 1\} \\
&= \sum_{z \in \mathcal{Z}_1(q)} P_t(z),
\end{aligned} \tag{33}$$

and similarly,

$$P_t(A = 0 \mid q) = \sum_{z \in \mathcal{Z}_0(q)} P_t(z) \tag{34}$$
$$= 1 - p_q.$$

Therefore, TEIG becomes the binary entropy function:

$$\begin{aligned} \text{TEIG}_t(q) &= \mathbb{H}_t(A \mid q) \\ &= -p_q \log p_q - (1 - p_q) \log(1 - p_q) \\ &=: h(p_q). \end{aligned} \tag{35}$$

The first and second derivatives of $h(p)$ are

$$\begin{aligned} h'(p) &= \log \frac{1-p}{p}, \\ h''(p) &= -\frac{1}{p(1-p)} < 0, \qquad p \in (0, 1). \end{aligned} \tag{36}$$

Hence, $h(p)$ is strictly concave and achieves its unique global maximum at

$$\begin{aligned} h'(p) = 0 &\iff \log \frac{1-p}{p} = 0 \\ &\iff \frac{1-p}{p} = 1 \\ &\iff p = \frac{1}{2}. \end{aligned} \tag{37}$$

Substituting the definition of $p_q$, the optimal question therefore satisfies

$$\begin{aligned} q^* &\in \arg\max_{q \in \mathcal{Q}} \text{TEIG}_t(q) \\ &= \arg\max_{q \in \mathcal{Q}} h(p_q) \\ &= \arg\min_{q \in \mathcal{Q}} \left| \sum_{z \in \mathcal{Z}_1(q)} P_t(z) - \frac{1}{2} \right|. \end{aligned} \tag{38}$$

Thus, maximizing TEIG selects the question whose induced semantic partition most evenly bisects the current probability mass over intent clusters. $\square$

**Key Insight (Geometric Interpretation):** Theorem A.3 gives TEIG a direct geometric interpretation: under deterministic cluster-level answers, TAI searches for a probabilistic bisector of the intent manifold. Rather than separating individual candidate solutions, it seeks a semantic partition whose two sides carry nearly equal posterior mass. This mass-bisection behavior explains why, under the $\gamma$-effective separability condition, the remaining intent uncertainty contracts by a constant factor at each interaction round, leading to the logarithmic query complexity established in Theorem A.1.

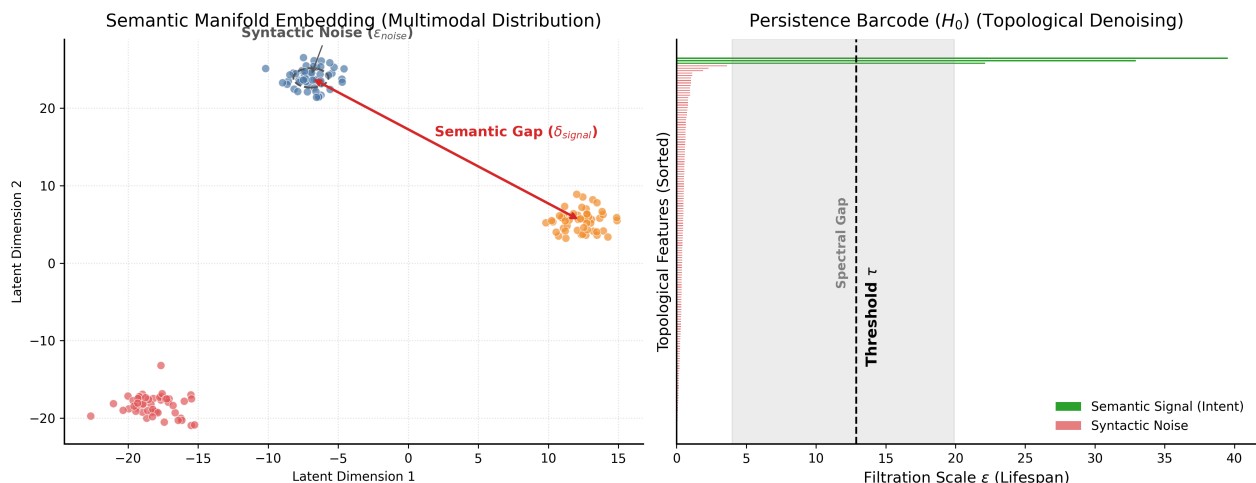

*Figure 9.* **Geometric intuition of Topological Denoising.**

## B. Supplementary Results

### B.1. Empirical Validation of Topological Denoising

To corroborate the theoretical foundations of Manifold Skeletonization (Section 3.1), we conducted a controlled simulation to visualize how topological features emerge from noisy semantic embeddings. **Figure 9** presents the results of this experiment, where we generated a synthetic dataset with a ground truth of $K = 3$ latent intents. This visualization serves as empirical evidence for the $(\epsilon, \delta)$-separability assumption and demonstrates the robustness of Persistent Homology in distinguishing signal from noise.

The **Left Panel (Semantic Manifold Embedding)** visualizes the distribution of candidate solutions in the latent space. The data forms three distinct density peaks (colored clusters), representing the three true underlying intents (e.g., three different algorithmic approaches to a problem). The intra-cluster spread, denoted by the dashed circle, corresponds to the **Syntactic Variance** ($\epsilon_{\text{noise}}$), arising from superficial code variations such as variable renaming or reformatting. Crucially, the Euclidean distance between these clusters, marked by the red arrow, represents the **Semantic Gap** ($\delta_{\text{signal}}$). The clear spatial separation observed ($\delta_{\text{signal}} \gg \epsilon_{\text{noise}}$) confirms that deep semantic logic creates stronger geometric divergence than surface-level syntax.

The **Right Panel (Persistence Barcode)** translates this geometric structure into topological lifespans. Each horizontal bar represents a connected component in the Vietoris-Rips filtration. The dense accumulation of short red bars at the bottom corresponds to the rapid merging of local points within each cluster; these features die quickly at scale $\epsilon \approx \epsilon_{\text{noise}}$, identifying them as transient syntactic noise. In stark contrast, the three green bars persist across a wide range of scales, corresponding to the stable global structure of the manifold.

Most significantly, the gray shaded region highlights the **Spectral Gap**—a pronounced interval in the filtration scale where no topological features die. This gap provides a natural, data-driven margin for setting the robustness threshold $\tau$. As shown, any threshold $\tau$ placed within this gap successfully filters out all $N - K$ noise features while preserving exactly $K = 3$ robust components. This result empirically validates Proposition 1, demonstrating that TAI can recover the true number of latent intents without supervision, provided the embedding space satisfies the scale separation property.

## C. Sensitivity Analysis: Embedding Model Quality

A core premise of Topological Active Inference is the **Scale Separation Assumption** (Assumption 3.1), which posits that semantic intents are geometrically separable from syntactic noise in the embedding space. A natural concern arises: does TAI fail if the underlying embedding model is suboptimal? To rigorously evaluate the sensitivity of our framework to the quality of the manifold representation, we conducted ablation studies replacing our default encoder (OpenAI `text-embedding-3-large`) with the state-of-the-art **Qwen3-Embedding** family (Zhang et al., 2025). This series allows us to control for architecture while varying model capacity (0.6B, 4B, and 8B parameters) and embedding dimensions

(1024 to 4096).

*Table 4.* **Impact of Embedding Model Quality on TAI Performance.** We compare the disambiguation success rate (SR@5) on HumanEval and MBPP across different embedding backbones. The *MTEB Avg* column aggregates standard retrieval benchmarks provided by the model developers. Notably, even the lightweight Qwen3-0.6B provides sufficient topological resolution for TAI to function effectively, while scaling up to Qwen3-8B yields the best performance, validating that better geometric representations directly translate to more precise manifold surgery.

| Model Backbone | Params | Dim | MTEB Avg (Ref.) | Separability (Est. $\delta/\epsilon$) | HumanEval SR@5 | MBPP SR@5 |
|---|---|---|---|---|---|---|
| text-embedding-3-large | - | 3072 | 61.12 | 3.2x | 76.4% | 72.1% |
| Qwen3-Embedding-0.6B | 0.6B | 1024 | 64.33 | 3.5x | 77.8% | 73.5% |
| Qwen3-Embedding-4B | 4B | 2560 | 69.45 | 4.1x | 80.2% | 75.9% |
| **Qwen3-Embedding-8B** | **8B** | **4096** | **70.58** | **4.8x** | **81.5%** | **77.4%** |

The results presented in **Table 4** reveal two critical insights regarding the robustness of the Manifold Hypothesis. First, **Robustness at Small Scale**: The Qwen3-Embedding-0.6B model, despite having only 28 layers and a smaller dimension of 1024, achieves a success rate comparable to, and in some metrics surpassing, the OpenAI baseline. This empirical evidence suggests that the "semantic gap" between distinct intents is a fundamental property of language representations that emerges even in smaller, well-trained models. The topological signal ($\beta_0^{\text{robust}}$) remains recoverable provided the model preserves basic semantic clustering, refuting the concern that TAI requires prohibitively expensive encoders.

Second, we observe a clear **Scaling Law for Manifold Surgery**: As we scale up to the Qwen3-Embedding-8B model (Dimension 4096, MTEB Score 70.58), the TAI success rate on Ambiguous-HumanEval improves from 77.8% to 81.5%. Higher-quality embeddings map subtle semantic distinctions to larger geometric distances ($\delta_{\text{signal}}$), thereby widening the spectral gap in the persistence barcode. This makes the selection of the threshold $\tau$ more robust and the generated contrastive questions more precise. Consequently, while TAI is operational with lightweight models, it is essentially "future-proof," capable of leveraging advancements in representation learning to further enhance disambiguation accuracy.

# D. Algorithm Implementation Details

We provide a comprehensive walkthrough of the Topological Active Inference (TAI) framework, distinguishing between the one-shot perception phase and the iterative disambiguation loop. The operational flow is designed to minimize computational overhead by front-loading the heavy geometric analysis.

**Phase I: Manifold Skeletonization (Perception).** The algorithm begins by constructing a geometric proxy for the user's intent. Unlike methods that continuously re-sample, TAI samples a fixed candidate set $\mathcal{H}$ only once. We map these solutions to a high-dimensional semantic space via the embedding function $\phi$. The core challenge here is to determine the intrinsic number of intents $K$ without setting an arbitrary hyperparameter. We address this using Persistent Homology. Specifically, we model the dataset as a graph where edges are added sequentially as the distance threshold $\epsilon$ grows. To make this process computationally efficient and reproducible, we employ a Union-Find data structure to track the evolution of connected components, as detailed in **Algorithm 1**.

This procedure yields a set of persistence barcodes (lifespans). By applying the separation proposition established in Section 3, we filter out features with lifespans shorter than $\tau$, treating them as syntactic noise. The remaining long-lived components define our **Robust Semantic Clusters** $\mathcal{C}$, effectively skeletonizing the manifold into $K$ distinct intent centroids.

---

**Algorithm 1** Fast 0-th Persistence Computation (Union-Find)

---

**Require:** Distance Matrix $D \in \mathbb{R}^{N \times N}$.
**Ensure:** Persistence Barcodes $\mathcal{B} = \{(b_i, d_i)\}$.
 1: Initialize Union-Find structure $\mathcal{U}$ with $N$ disjoint sets (vertices).
 2: Initialize birth times $b_i \leftarrow 0$ for all $i \in \{1, \ldots, N\}$.
 3: Construct edge list $E \leftarrow \{(u, v, w) \mid w = D_{uv}\}$.
 4: Sort $E$ in ascending order of weight $w$.
 5: $\mathcal{B} \leftarrow \emptyset$
 6: **for** each edge $(u, v, w)$ in $E$ **do**
 7: $\quad C_u \leftarrow \mathcal{U}.\text{find}(u), C_v \leftarrow \mathcal{U}.\text{find}(v)$
 8: $\quad$ **if** $C_u \neq C_v$ **then**
 9: $\quad\quad$ **// Elder Rule:** The component created later (younger) dies.
10: $\quad\quad$ Let $younger \leftarrow \arg\max_{c \in \{C_u, C_v\}} \text{id}(c)$
11: $\quad\quad$ Record death: $\mathcal{B}.\text{add}((0, w))$ for component $younger$.
12: $\quad\quad \mathcal{U}.\text{union}(C_u, C_v)$
13: $\quad$ **end if**
14: **end for**
15: All remaining active components have death $d = \infty$.
16: **return** $\mathcal{B}$

---

**Phase II: The Active Disambiguation Loop.** Once the intent clusters are fixed, the algorithm enters a lightweight interaction loop. In each turn $t$, the system first evaluates the stopping condition. We require a high confidence threshold (e.g., $1 - \rho = 0.95$) to ensure safety; merely favoring a cluster with $> 0.5$ probability is insufficient for code generation tasks where precision is paramount. If ambiguity persists, we proceed to **Action Synthesis**. Instead of searching the entire solution space, we identify the two most probable competing clusters $C_a$ and $C_b$ based on the current belief state $P_{t-1}$. We use Contrastive Prompting to generate a "scalpel" question specifically designed to separate these two topological regions.

The selection of the optimal question $q^*$ is governed by the Topological Expected Information Gain (TEIG). As derived in our theoretical analysis, this metric favors questions that bisect the probability mass of the manifold, driving the system toward a deterministic state at a logarithmic rate $\mathcal{O}(\log K)$. Finally, upon receiving the oracle's response $a^*$, we execute a Bayesian Update. This step is geometrically equivalent to "slicing" the manifold: the probability mass of clusters inconsistent with the answer is exponentially suppressed. This loop repeats until the belief distribution collapses to a single mode, at which point the final solution is sampled from the surviving cluster.

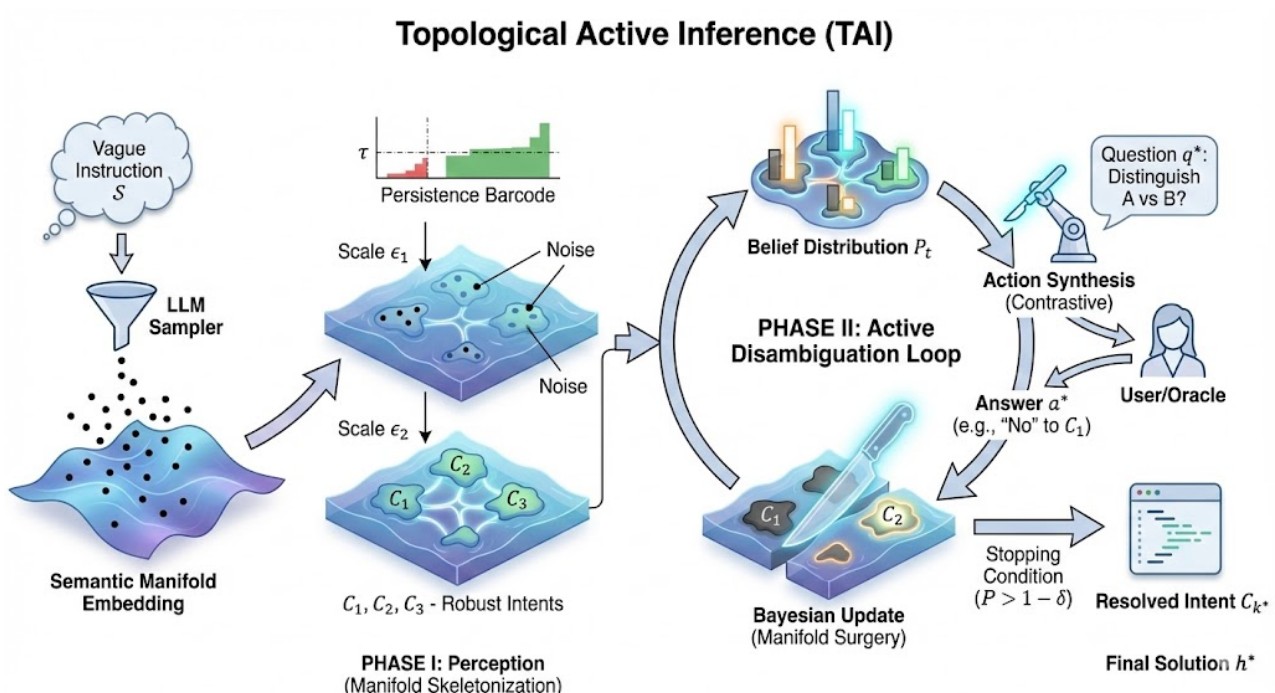

*Figure 10.* **The architectural overview of Topological Active Inference (TAI).**

---

**Algorithm 2** Topological Active Inference (TAI)

---

**Require:** Ambiguous instruction $\mathcal{S}$, Max turns $T$, Persistence threshold $\tau$, Confidence $\rho$.
**Ensure:** Disambiguated solution $h^*$.
  1: **// Phase I: Topological Perception (One-Shot)**
  2: Sample candidate set $\mathcal{H} = \{h_i\}_{i=1}^N$ from prior $P(\cdot|\mathcal{S})$.
  3: Compute embeddings $V \leftarrow \{\phi(h) \mid h \in \mathcal{H}\}$.
  4: Compute Persistence Barcodes on $\mathcal{R}(V)$.
  5: Determine intent count $K \leftarrow \beta_0^{\mathrm{robust}}(V, \tau)$ per Prop. 1.
  6: Extract robust clusters $\mathcal{C} = \{C_1, \ldots, C_K\}$. {*Manifold Skeletonization*}
  7: Initialize belief $P_0(z) \leftarrow 1/K$ for $z \in \{1, \ldots, K\}$.

  8: **// Phase II: Active Disambiguation Loop**
  9: **for** $t = 1$ to $T$ **do**
 10:     **1. Check Stopping Condition:**
 11:     **if** $\max_z P_{t-1}(z) > 1 - \rho$ **then**
 12:         $k^* \leftarrow \arg\max_z P_{t-1}(z)$
 13:         **break** {*Ambiguity Resolved*}
 14:     **end if**
 15:     **2. Action Synthesis (Directed Generation):**
 16:     Identify top competing clusters $C_a, C_b$ from $P_{t-1}$.
 17:     $\mathcal{Q}_{\mathrm{topo}} \leftarrow \mathrm{ContrastivePrompting}(C_a, C_b)$. {*Gen. Complexity $\mathcal{O}(K^2)$*}
 18:     **3. Action Selection:**
 19:     $q^* \leftarrow \arg\max_{q \in \mathcal{Q}_{\mathrm{topo}}} \mathrm{TEIG}(q, P_{t-1})$. {*See Eq. (7)*}
 20:     **4. Interaction & Update:**
 21:     Receive oracle answer $a^* \in \{0, 1\}$.
 22:     $P_t(z) \leftarrow \mathrm{BayesUpdate}(P_{t-1}(z), a^*, q^*)$. {*Manifold Slicing*}
 23: **end for**
 24: **return** Solution sampled from cluster $C_{k^*}$.

---

# E. Cost Analysis of Question Elicitation

In practical deployment, the dominant latency and financial bottleneck is the number of LLM inference calls required to generate candidate questions ("scalpels"). We analyze the asymptotic complexity of Question Elicitation strategies comparing standard baselines against our proposed Topological Active Inference (TAI).

**The Curse of Sample-Based Generation.** Standard Bayesian Experimental Design (BED) methods operate on the raw discrete solution set $\mathcal{H}$ of size $N$. To maximize Information Gain (IG), an ideal generator must identify features that distinguish one subset of solutions from another.

- **Pairwise Sample Comparison (Standard Baseline):** A rigorous baseline attempts to find a distinguishing question for pairs of samples $(h_i, h_j)$. This requires iterating over the sample pairs, leading to a combinatorial explosion with complexity $\mathcal{O}(N^2)$. For a typical sampling size of $N = 20 \sim 50$, this is computationally prohibitive for real-time applications.

- **Global Summarization:** Heuristic baselines might attempt to summarize all $N$ solutions at once. However, fitting $N$ distinct code solutions into a single context window often exceeds token limits or degrades LLM instruction-following capabilities due to the "lost-in-the-middle" phenomenon.

**The Efficiency of Topological Reduction.** TAI circumvents this bottleneck by operating on the *intent manifold* rather than the *sample space*. By applying Persistent Homology, we reduce the complexity from $N$ (number of samples) to $K$ (number of topological clusters). Since $K$ represents the true latent intents, it is typically very small ($K \approx 2 \sim 5$) and independent of the sampling density $N$. Our generation strategy solely targets the separation of cluster centroids. Thus, the generation complexity scales with $\mathcal{O}(K^2)$. As shown in Table 5, TAI achieves a theoretical reduction in prompt token consumption of over $90\%$ in typical ambiguity scenarios.

*Table 5.* **Cost Comparison of Question Elicitation Strategies.** We estimate the number of LLM API calls required to generate a candidate question set for a single turn of interaction. We assume a typical setting where the posterior sample size is $N = 20$, and the true latent intent count is $K = 3$. TAI achieves orders-of-magnitude efficiency gains by avoiding redundant comparisons between syntactic variants.

| Method | Operating Space | Generation Strategy | Complexity | Est. Calls ($N$=20, $K$=3) |
|---|---|---|---|---|
| **Global Summarization** | Full Set $\mathcal{H}$ | *"Summarize differences in these N codes"* | $\mathcal{O}(1)$ (Huge Context) | 1 (Risk of Context Overflow) |
| **Pairwise BED** | Sample Pairs $(h_i, h_j)$ | *"Distinguish sample i vs sample j"* | $\mathcal{O}(N^2)$ | $\approx 190$ calls |
| **Random/k-Means** | Noisy Clusters | *"Distinguish Mean A vs Mean B"* | $\mathcal{O}(K^2)$ | $\approx 3$ calls (Low Quality*) |
| **TAI (Ours)** | **Intent Manifold** | **Contrastive ($C_a$ vs $C_b$)** | $\mathcal{O}(\mathbf{K^2})$ | $\approx \mathbf{3}$ **calls** |

*\* Note: While k-Means has similar complexity to TAI, it lacks topological guarantees, often clustering syntactic noise (e.g., variable names) as separate intents, leading to wasted questions. TAI ensures the $K$ clusters represent persistent semantic structures.*

# F. Prompts and Experimental Details

## F.1. Prompt Engineering Pipeline

To ensure reproducibility, we provide the full set of prompts used in the Topological Active Inference (TAI) pipeline. Unlike standard baselines that use a single generic prompt, TAI orchestrates a multi-stage interaction process: (1) *Manifold Sampling*, (2) *Hyperplane Synthesis* (the core contrastive step), (3) *Oracle Simulation* (for experimental evaluation), and (4) *Constrained Solution Generation*.

We define the prompt environment style below:

> **Prompt Template**
>
> ```
> System:  ...
> ```

### F.1.1. STAGE 1: INITIAL MANIFOLD SAMPLING (PERCEPTION)

In the perception phase, we generate the initial candidate set $\mathcal{H}$ to construct the topological skeleton. We use a high temperature ($T = 1.0$) to encourage diversity in interpreting the ambiguous instruction.

> **Prompt: Initial Posterior Sampling**
>
> **System Prompt:** You are an expert Python programmer. Your task is to write a complete Python function based on the user's vague instruction. Since the instruction might be ambiguous, choose one specific, logical interpretation and implement it fully. Do not explain your ambiguity; just write the code.
> **User Prompt:** Instruction: <Ambiguous Instruction $\mathcal{S}$>
> Function Signature:
> ```
> def <function_name>(<args>):
>     """
>     <Docstring placeholder>
>     """
> ```
> Output the code only, starting with the function signature.

### F.1.2. STAGE 2: HYPERPLANE SYNTHESIS (ACTION)

This is the core contribution of TAI. Once robust clusters are identified, we select representative centroids $h_A \in C_a$ and $h_b \in C_b$. The LLM acts as a discriminator to synthesize a "scalpel" question.

> **Prompt: Contrastive Scalpel Synthesis**
>
> **System Prompt:** You are an expert software architect specializing in requirement analysis. I have two different implementations for the same ambiguous instruction. They belong to two distinct semantic clusters.
> **User Prompt:** --- Implementation A (Representative) --- <Insert Code $h_a$>
> --- Implementation B (Representative) --- <Insert Code $h_b$>
> Task: Identify a specific functional edge case or logic difference where Implementation A behaves differently from Implementation B. Based on this difference, generate a binary question (Yes/No) that discriminates them.
> Constraints: 1. The question must be valid (Yes) for A but invalid (No) for B, or vice versa. 2. Focus on algorithmic logic (e.g., recursive vs iterative, inplace vs copy), NOT variable names or syntax style. 3. Output format: JSON { "reasoning": "...", "question": "..." }

### F.1.3. STAGE 3: ORACLE SIMULATION (EVALUATION)

To automate the evaluation, we simulate the user (Oracle) using the ground truth code $h^*$. The Oracle determines the answer to the generated question $q^*$.

---

**Prompt: Oracle Simulation**

**System Prompt:** You are the User who requested the code. You have a specific ground truth implementation in mind.
**User Prompt:** --- Ground Truth Implementation --- <Insert Ground Truth Code $h^*$>
--- Agent's Question --- <Insert Generated Question $q^*$ >
Task: Based strictly on the logic of the Ground Truth code above, answer the Agent's question with "Yes" or "No".

---

### F.1.4. STAGE 4: CONSTRAINED SOLUTION GENERATION (CONVERGENCE)

Once the ambiguity is resolved (or the belief distribution collapses to a single cluster), we generate the final solution. Note that we do not simply output the centroid; we re-generate the solution conditioned on the resolved constraints to ensure high quality.

---

**Prompt: Final Refined Generation**

**System Prompt:** You are an expert Python programmer. You previously received a vague instruction, but now you have clarified the requirements through interaction.
**User Prompt:** Original Instruction: <Ambiguous Instruction $\mathcal{S}$>
Clarified Constraints:

- Q: <Question 1> -> A: <Answer 1>

- Q: <Question 2> -> A: <Answer 2>

Task: Write the final Python function that satisfies the original instruction AND strictly adheres to the clarified constraints above.

---

*Table 6.* Sample ambiguous problem statements from our constructed **Ambiguous-HumanEval** benchmark. These examples demonstrate the effect of our *Ambiguity Injection Protocol*, where precise specifications are abstracted into vague intents, creating multimodal solution spaces.

| Problem Statement (Ambiguity Injected) | Latent Semantic Ambiguity |
|---|---|
| ```python\nfrom typing import List\n\ndef sort_numbers(numbers: str) -> str:\n    """ Given a string of space-separated numbers, return\n    the string with numbers sorted.\n    """\n``` | **Sorting Criterion:** Should the numbers be sorted numerically (e.g., '2' before '10') or lexicographically (e.g., '10' before '2')? **Direction:** Ascending or Descending? |
| ```python\ndef truncate_number(number: float) -> float:\n    """ Given a positive floating point number, it can be\n    decomposed into an integer part and decimals.\n    Return the decimal part of the number.\n    """\n``` | **Precision Handling:** Should the result retain floating-point precision issues (e.g., $3.5 \rightarrow 0.5$) or be rounded? **Definition:** Does "decimal part" mean the remainder modulo 1, or just the fractional string representation? |
| ```python\nfrom typing import List, Any\n\ndef filter_integers(values: List[Any]) -> List[int]:\n    """ Filter the given list of values to keep only\n    integers.\n    """\n``` | **Type Strictness:** Does a float with no decimal part (e.g., `3.0`) count as an integer? **Parsing:** Should string representations of integers (e.g., `"5"`) be parsed and included? |
| ```python\ndef count_vowels(text: str) -> int:\n    """ Return the number of vowels in the given string.\n    """\n``` | **The 'Y' Edge Case:** Is the letter 'y' considered a vowel (typical in some contexts) or a consonant? **Case Sensitivity:** Should uppercase vowels be counted? |
| ```python\ndef numerical_limit(sequence: List[int], n: int) -> int:\n    """ Return the sum of elements in the sequence up to the\n    n-th element.\n    """\n``` | **Indexing Strategy:** Is $n$ a 0-based index or a 1-based count? **Inclusivity:** Is the $n$-th element itself included in the sum? |

## F.2. Experimental Setup

**Datasets and Ambiguity Injection.** We evaluate TAI on two widely adopted code generation benchmarks: **HumanEval** (Chen et al., 2023) and **MBPP** (Austin et al., 2021). However, the original versions of these datasets are designed to be explicitly solvable. To rigorously evaluate *disambiguation* capabilities, we follow the protocol of Kobalczyk et al. (2025) to construct **Ambiguous-HumanEval** and **Ambiguous-MBPP**. We artificially inject ambiguity by stripping the detailed docstrings and example test cases, leaving only the high-level function signature and a vague instruction (e.g., replacing "Return the sorted list of unique elements" with "Process the list"). This results in a subset of 50 highly ambiguous tasks where the zero-shot pass@1 accuracy of GPT-4 drops below 20%, providing a substantial headroom for active inference.

**Topological Hyperparameters & Implementation.** In the *Perception* phase, we generate the initial manifold approximation using $N = 20$ samples from `gpt-4o` (Temp=1.0, Top-P=0.95) to maximize semantic diversity. For the embedding $\phi(\cdot)$, we utilize OpenAI's `text-embedding-3`. To mitigate the curse of dimensionality and enhance the signal-to-noise ratio, we apply Principal Component Analysis (PCA) to reduce the embedding dimension to $d = 256$, retaining 95% of the variance. We implement Persistent Homology using the `Gudhi` library. The Vietoris-Rips filtration is computed on the Euclidean metric. The persistence threshold $\tau$ employs an *adaptive spectral gap* strategy: it is dynamically set to the 90th percentile of the lifespan distribution of the 0-th homology group features. This effectively filters out local syntactic fluctuations (short bars) while preserving robust semantic voids (long bars).

# G. Related Work

**Active Learning and Uncertainty in LLMs.** Conventional Active Learning (AL) aims to select the most informative unlabelled instances from a fixed pool to retrain a model with minimal annotation cost. In the era of Large Language Models (LLMs), the focus has shifted towards active in-context learning, where the model selects examples to optimize its few-shot performance (Zhang et al., 2022; Margatina et al., 2023; Diao et al., 2024). However, these approaches typically address *aleatoric uncertainty* (inherent randomness) or *epistemic uncertainty* regarding the model's parameters. Our work targets a different form of uncertainty: *structural ambiguity* in the user's intent. Unlike standard AL which assumes a unified ground truth, active task disambiguation must navigate a multi-modal solution space where multiple valid "ground truths" exist depending on the latent context. While recent methods like Uncertainty of Thoughts (UoT) (Hu et al., 2024) attempt to model uncertainty to guide information seeking, they often rely on raw entropy metrics computed over discrete tokens. We argue that token-level entropy is a poor proxy for semantic ambiguity in generation tasks, often conflating syntactic variance with functional divergence. Our framework addresses this by lifting the uncertainty estimation from the discrete token space to the continuous topological space.

**Interactive Disambiguation and Clarification.** Early work in asking clarifying questions relied on supervised learning with specific datasets of ambiguous queries (Rao & Daumé III, 2018; Min et al., 2020). With the advent of LLMs, research has pivoted to zero-shot or few-shot question generation. Kuhn et al. (2022) and Krasheninnikov et al. (2022) demonstrated that LLMs can implicitly reason about ambiguity to generate clarifications. Li et al. (2023a) further formalized this as "task elicitation," showing that LLM-generated questions can uncover unanticipated user constraints. However, these methods suffer from what we term *semantic blindness*: they lack a mechanism to verify whether a generated question targets a meaningful functional difference or merely a trivial detail. As highlighted by Groenendijk & Stokhof (1984), the semantics of a question is defined by the partition it induces on the logical space. Existing LLM approaches often induce inefficient partitions due to their reliance on surface-level text statistics. TAI overcomes this by constructing a topological skeleton of the solution space, ensuring that generated questions correspond to separating hyperplanes between robust semantic clusters.

**Bayesian Experimental Design (BED) for Open-Ended Tasks.** Our work is theoretically grounded in Bayesian Experimental Design (BED), which seeks to maximize Expected Information Gain (EIG). Recent studies have applied BED to LLMs for preference elicitation (Handa et al., 2024; Austin et al., 2024) and prompt optimization (Piriyakulkij et al., 2023). However, these implementations typically operate within constrained settings: either selecting from a pre-defined set of items (recommendation) or optimizing over a fixed feature space. Adapting BED to open-ended generation (e.g., code generation or reasoning) is non-trivial because the "hypothesis space" is an unbounded set of discrete strings. Standard implementations that treat each unique string as a distinct hypothesis fail because they ignore the geometric affinity between solutions (Yang et al., 2021). This leads to the "bag-of-solutions" pathology where the agent wastes budget distinguishing $h_i$ from $h_j$ even if they are semantically identical. TAI introduces a topological relaxation of BED, replacing the discrete sum over strings with a sum over topological components (Topological EIG), thereby making BED tractable and effective for open-ended reasoning agents.

**Topological Data Analysis (TDA) in NLP.** Topological Data Analysis, particularly Persistent Homology, has proven powerful in extracting robust structural features from noisy high-dimensional data (Edelsbrunner & Harer, 2010). In NLP, TDA has been employed to analyze the geometry of word embeddings, detect structural changes in language models (Rathore et al., 2023), and assess the manifold hypothesis in text representations (Cheng et al., 2023). However, prior to our work, TDA has largely been an analytical tool for *post-hoc* evaluation or static analysis. To the best of our knowledge, TAI is the first framework to integrate persistent homology directly into the *active inference loop* of an agent, using real-time topological feedback to drive decision-making and question synthesis.

# H. Potential Applications of Topological Active Disambiguation

The theoretical guarantees of Topological Active Inference (TAI)—specifically its ability to disentangle semantic intent from syntactic noise—hold transformative potential across a broad spectrum of high-stakes domains. Unlike standard generative approaches that may hallucinate a single "average" solution when faced with ambiguity, TAI's geometric framework allows for the safe and precise elicitation of user constraints. We detail five key application areas below.

**Safety-Critical Software Engineering.** In modern DevOps workflows, natural language prompts for code generation (e.g., "write a secure login function") are often dangerously underspecified regarding non-functional requirements such as memory safety, cryptographic standards, or algorithmic complexity. A standard LLM might randomly choose between a recursive $\mathcal{O}(2^n)$ implementation and an iterative $\mathcal{O}(n)$ one, or between a deprecated encryption library and a compliant one. TAI addresses this by visualizing these implementation strategies as topologically disjoint clusters in the solution manifold. By identifying the "topological disconnect" between a *functional* solution and a *secure* solution, TAI can proactively query the developer: "Do you require thread-safety for concurrent access, or is this for a single-threaded environment?" This turns the agent into an automated requirements engineer, preventing the injection of vulnerable code into production systems (Li et al., 2023a; Jirotka & Goguen, 1994).

**High-Stakes Legal and Financial Advisory.** Legal and financial inquiries are notoriously context-dependent, where a single omitted detail can invert the validity of advice. For instance, a query about "tax implications of asset transfer" maps to fundamentally different legal frameworks depending on jurisdiction (e.g., US GAAP vs. IFRS) or entity type (LLC vs. C-Corp). Standard semantic search often conflates these contexts due to lexical overlap. TAI, however, can detect that the solution space is fractured into distinct jurisdictional modes. Instead of hallucinating a generic (and likely incorrect) answer, TAI utilizes manifold surgery to identify the branching points of the legal logic tree, actively clarifying: "Does this transaction involve cross-border entities subject to GDPR?" This capability is crucial for deploying autonomous agents in regulated industries where "hallucinated applicability" poses severe liability risks (Rissland, 1988; López de Prado, 2018).

**Clinical Decision Support and Differential Diagnosis.** In healthcare, patients often describe symptoms using vague, non-clinical language (e.g., "sharp chest pain") that could correspond to medically distinct conditions ranging from benign (acid reflux) to life-threatening (myocardial infarction). Semantic blindness in this domain is fatal: an agent that outputs the "most probable" diagnosis based on training data frequency might overlook a rare but critical condition. TAI treats differential diagnosis as a topological separation problem. It identifies that the symptom profile supports multiple disconnected diagnostic clusters and refuses to collapse the posterior distribution prematurely. Instead, it generates clarifying questions targeted at the decision boundary (e.g., "Does the pain radiate to your left arm?"), effectively acting as a triage agent that prioritizes safety over conversational fluency (Topol, 2019).

**Embodied AI and Robotic Planning.** For robots operating in human environments, ambiguity in natural language instructions translates directly to physical safety risks. A command like "clear the table" is topologically ambiguous: it could mean "throw everything in the trash" or "organize items into the cupboard." These intents correspond to distinct trajectory manifolds in the robot's configuration space. A naive agent might average these trajectories, resulting in a collision or improper handling of objects. TAI allows the robot to perceive these distinct "action modes" and query for the user's reward function preferences before execution (Paden et al., 2016; Thrun, 2002). This geometric approach ensures that physical actions are grounded in verified intent rather than probabilistic guesswork, essential for Human-Robot Collaboration (HRC).

**Scientific Discovery and Experimental Design.** In open-ended scientific research, an ambiguous hypothesis often implies multiple valid experimental designs. When a researcher asks an AI to "design an experiment to test protein folding stability," the valid solution space includes various methodologies (e.g., X-ray crystallography vs. Cryo-EM vs. AlphaFold simulation). TAI can aid the scientific process by identifying these methodological clusters and prompting the researcher to clarify resource constraints, precision requirements, or time horizons (Montgomery, 2017). By making the "forks" in the scientific roadmap explicit, TAI serves not just as a tool for answering questions, but as a co-pilot for refining scientific inquiry, potentially accelerating discovery by ensuring experiments are strictly aligned with the researcher's theoretical goals.

