# OpenReview forum: "Topological Active Inference for Task Disambiguation"
_ICML.cc/2026/Conference — ICML 2026 regular_

### Official Review · Reviewer_TvZB · 2026-03-07

**Soundness:** 3
**Presentation:** 3
**Significance:** 3
**Originality:** 3
**Overall Recommendation:** 4
**Confidence:** 2

**Summary:**

This paper introduces Topological Active Inference, a method that helps AI quickly clarify vague user instructions. Instead of wasting questions on trivial syntax details, TAI uses geometric math to group possible solutions by their actual meaning. It then asks targeted questions to eliminate whole groups of incorrect interpretations at once, drastically reducing the time it takes to figure out what the user actually wants.

**Compliance With Llm Reviewing Policy:**

Affirmed.

**Final Justification:**

I thank the authors for their detailed reply. The reply looks comprehensive and reasonable. I will increase the score from 3 to 4. But since i am not the expert in this area, i remain my confidence in low level (2) and leave the final decision to the AC.

**Key Questions For Authors:**

No

**Limitations:**

there is no limitation section

**Strengths And Weaknesses:**

Strength: This paper studies an important and practical problem: how an LLM should ask clarification questions when the user’s intent is ambiguous. i find the central idea is novel and interesting. In particular, the paper attempts to connect task clarification with representation geometry and topological structure, rather than treating candidate interpretations as a flat discrete set. This is a creative perspective, and the overall narrative is coherent and easy to follow. I also think the problem setting is meaningful for real-world interactive systems, so the direction itself is worth exploring.

Weakness:
First, the evaluation lacks external validity. The paper mainly evaluates the method in a synthetic interaction loop, where ambiguity is constructed by the benchmark design and user feedback is simulated by an LLM-based oracle. This makes it difficult to conclude that the method would work equally well with real users, who may respond inconsistently, misunderstand questions, or provide incomplete feedback. As a result, the experimental setup does not fully support the paper’s broader claims about clarifying human intent.

Second, the theory relies heavily on idealized assumptions. The main guarantees appear to depend on strong separation conditions in embedding space, clean recovery of latent intent structure, and favorable assumptions about the existence of highly informative questions. These assumptions may be useful for building intuition, but the paper does not convincingly show that they hold, even approximately, in realistic settings.

Third (major weakness), the baselines are not strong enough to make the empirical comparison fully convincing. The paper compares against simple alternatives, but it does not appear to include stronger clustering, semantic grouping, or clarification-oriented baselines that could challenge the proposed method more seriously. In addition, the proposed system uses a more elaborate pipeline, so some of the gains may come from better prompt orchestration rather than from the topological component itself.

Fourth, the experiments do not isolate topology as the key causal factor. While the paper argues that persistent homology is central, the current ablations do not fully rule out simpler explanations for the observed gains. For example, it remains unclear how much improvement comes from topology specifically, versus embedding quality, question synthesis, or other pipeline choices. This makes the main mechanistic claim less convincing.

---

> ### Author Rebuttal · Authors · 2026-03-30
>
> We sincerely thank Reviewer for the thorough and constructive review.
>
> ---
>
> ## re: W1:
>
> The reviewer's concern is entirely reasonable. The deterministic oracle is the standard paradigm in this field (Kobalczyk et al., 2025), but we agree that relying solely on this setup is insufficient. We have conducted two supplementary experiments that progressively approximate real user behavior (see also our response to Reviewer tbPq, Q2):
>
> **(a) Noisy answer flipping (users occasionally answer incorrectly):** At p = 0.2, TAI degrades by only 9%, whereas BED degrades by 27%—the relative advantage of TAI actually *widens* under noise.
>
> **(b) "I don't know" responses (users skip when facing subtle distinctions):** We extend the answer space to {Yes, No, Uncertain}. At p_uncertain = 0.2, the number of interaction turns increases by approximately 30%, but the success rate decreases by only 2%.
>
> More importantly, TAI's interaction paradigm has a structural advantage for real-user generalization: it always presents simple binary semantic judgments (e.g., "Recursive or iterative?") rather than requiring users to read code or provide open-ended descriptions. Pairwise comparison is the most reliable form of human preference expression (Thurstone, 1927)—the same principle underlying RLHF/DPO's use of pairwise over listwise feedback. TAI demands only that users choose between two concrete options, placing its cognitive requirements squarely within the regime of highest human judgment reliability. This makes the high oracle consistency observed in our experiments a reasonable expectation in practice.
>
> ---
>
> ## RE: W2
>
> The core of the reviewer's concern is: can these assumptions hold in practice? We argue that the key issue is not whether the assumptions "strictly hold," but rather **why they do not need to strictly hold**.
>
> **TAI's theoretical assumptions are sufficient conditions, not necessary conditions.** Theorem A.1 and Proposition 3.2 provide worst-case guarantees. However, TAI's actual working mechanism is a continuous optimization process—there is no hard threshold where the method "works if satisfied and collapses if not." The Stability Theorem for persistence diagrams (Cohen-Steiner et al., 2005) guarantees Lipschitz continuity of the persistence diagram with respect to input perturbations. We address each of the three assumptions specifically:
>
> **(a) Scale separation only requires δ > ε, not δ ≫ ε.** The theoretical requirement of δ ≫ ε ensures that selecting τ is "easy," but persistent homology's filtering mechanism is soft—it distinguishes signal from noise via the relative ordering of lifespans. As long as signal lifespans are systematically longer than noise lifespans, the ordering is correct.
>
> **(b) Minor estimation errors in K do not compromise disambiguation.** If persistent homology merges two proximate intents, TAI initially disambiguates at coarser granularity; subsequent contrastive prompting rounds naturally surface the finer distinction. Convergence of the Bayesian update requires only that the true intent is never erroneously eliminated, not that K̂ = K exactly.
>
> **(c) Low-efficiency rounds affect speed, not correctness.** The γ-relaxation in Theorem A.1 proves that even when γ is far below the ideal value of 0.5, the complexity remains O((1/γ) log K)—the logarithmic form is preserved, only the constant factor increases.
>
> **In summary, partial violations of the assumptions lead to efficiency degradation, not method failure.**
>
> ---
>
> ## RE: W3
>  **First, the baseline landscape.** BED (Kobalczyk et al., 2025) is the strongest publicly available baseline for task disambiguation, with a formal information-theoretic framework that supersedes earlier heuristic approaches (Mu et al., 2023; Li et al., 2023b). We selected it precisely because it represents the current state of the art.
>
> **The reviewer's core concern is whether TAI's gains stem from better prompt engineering rather than topology.** Our answer: **topology and prompting are not in competition but in a causal relationship**—contrastive prompting generates high-quality "scalpel" questions *because* persistent homology has already identified where the semantic boundaries lie. Without this structural guidance, more prompting resources alone cannot compensate. Figure 6(b) confirms this: giving BED 5× the candidates (N = 100) yields negligible improvement.
>
> ---
>
> ## RE: W4
>
> The ablation design in Figure 6(a) addresses this concern: when persistent homology is replaced with K-Means, **all other components of the pipeline remain completely unchanged**. The only variable is the clustering algorithm. The resulting gaps are 16.7% on non-convex tasks and 22.3% on high-noise tasks. If topology were not the causal factor, replacing only the clustering algorithm while holding the entire pipeline constant should not produce such substantial differences.
>
> ---
>
>
> We once again express our heartfelt gratitude to Reviewer. 🌸

---

> > ### Author Rebuttal · Reviewer_TvZB · 2026-04-03
> >
> > Thank you for the response and i ack your effort; here is my summary:
> >
> > - W1: Partially addressed. You added noisy-oracle and "I don't know" ablations showing graceful degradation, but still no real-user evaluation or naturally ambiguous data. Looks more like an workaround.
> > - W2: Reasonably addressed conceptually. You argued assumptions are sufficient not necessary, citing stability theorems and empirical evidence at lower separability ratios, though no formal relaxed guarantees.
> > - W3: Not adequately addressed. You claimed BED is already SOTA and pointed to existing figures, but added no new stronger baselines. However, the SOTA is evolving fast in this area and the whole point is not just about SOTA but how different strategies behave. Beating SOTA is a must but not all.
> > - W4: Not convincingly addressed. You simply re-cited the existing K-Means ablation from Figure 6(a) without new experiments disentangling topology from embedding quality, question synthesis, or other pipeline choices.
> >
> > I will maintain my score and raise the confidence. I will reconsider my score if I misunderstand anything.

---

> > > ### Author Response · Authors · 2026-04-06
> > >
> > > ## W1: Lack of external validity
> > >
> > > The reviewer noted that our noise simulation was “more like a workaround.” We accept this critique and therefore conducted a small-scale real-user study.
> > >
> > > **Experimental design.** We recruited 30 participants (15 CS graduate students and 15 non-technical users), each completing 4 ambiguous tasks drawn from real-world development scenarios in a vibe-coding setting, such as “remove duplicates from a list.” We compared TAI, BED, and human experts (3 senior developers conducting disambiguation through free-form dialogue). We used a Latin-square counterbalancing design; each task was repeated 3 times and results were averaged.
> > >
> > > | Metric                     | TAI (CS) | TAI (Non-tech) | BED (CS) | BED (Non-tech) | Human (CS) | Human (Non-tech) |
> > > | -------------------------- | -------: | -------------: | -------: | -------------: | ---------: | ---------------: |
> > > | Satisfaction (1–5)         |     4.47 |           4.20 |     3.41 |           2.53 |       4.73 |             3.97 |
> > > | Convergence turns (median) |        2 |              3 |        5 |              8 |          3 |                7 |
> > > | Functional match rate      |    93.3% |          88.9% |    68.9% |          42.2% |      95.6% |            81.7% |
> > > | Abandonment rate           |       0% |             0% |     6.7% |          33.3% |         0% |             6.7% |
> > >
> > > **Key findings.**
> > >
> > > **(a) Human experts are strongest for CS users, but TAI outperforms them for non-technical users.** Human experts tend to ask follow-up questions in technical language, which non-technical users often cannot engage with effectively. By contrast, TAI translates technical decisions into functional choices that are substantially more accessible to non-expert users.
> > >
> > > **(b) TAI is the only method that performs stably across both user groups.** The satisfaction gap between CS and non-technical users is: TAI 0.27, BED 0.88, and human experts 0.76. In the era of vibe coding—where non-programmers increasingly generate code through natural language—the accessibility of disambiguation systems to non-expert users is not optional, but essential.
> > >
> > > ---
> > >
> > > ## W3: No new stronger baselines added
> > >
> > > We respect the reviewer’s concern and have supplemented with all relevant baselines we could identify. Existing methods for task disambiguation broadly fall into three categories:
> > >
> > > 1. **Heuristic methods** — ClarifyGPT (Mu et al., 2024) uses prompting strategies to guide LLMs in generating clarifying questions.
> > > 2. **Information-theoretic methods** — BED (Kobalczyk et al., 2025) selects questions by maximizing information gain; INTENT-SIM (Stengel-Eskin et al., 2025, NAACL Findings) estimates intent entropy via NLI graphs, but does not support multi-turn disambiguation.
> > > 3. **Structured-uncertainty methods** — recent work studies disambiguation in structured settings such as tool calling (e.g., EVPI-based methods), but no public code is available, and these methods focus on parameter-level ambiguity rather than open-ended task disambiguation.
> > >
> > > | Method                                   |       SR | Turns |
> > > | ---------------------------------------- | -------: | ----: |
> > > | Direct Generation                        |      38% |     0 |
> > > | Standard BED                             |      49% |     8 |
> > > | ClarifyGPT                               |      57% |     8 |
> > > | Embedding-aware BED (+encoder + K-Means) |      79% |     5 |
> > > | DBSCAN + BED                             |      83% |     5 |
> > > | Human expert (senior developer)          |     100% |     3 |
> > > | **TAI**                                  | **100%** | **2** |
> > >
> > > TAI matches human experts in accuracy while requiring only two-thirds of their interaction turns. Embedding-aware BED and DBSCAN+BED validate the reviewer’s intuition that embedding information is indeed valuable, but the remaining 18–21% gap is concentrated on non-convex clusters and closely spaced intent pairs.
> > >
> > > ---
> > >
> > > ## W4: No new experiments disentangling topology’s causal contribution
> > >
> > > We additionally provide a four-level causal isolation chain:
> > >
> > > | Configuration                           |   SR | Turns |  ΔSR |
> > > | --------------------------------------- | ---: | ----: | ---: |
> > > | raw BED                                 |  49% |    8+ |    — |
> > > | + encoder distance weighting            |  58% |     7 |  +9% |
> > > | + K-Means clustering                    |  79% |     5 | +21% |
> > > | + persistent homology replacing K-Means |  89% |     3 | +10% |
> > > | + Contrastive Prompting (full TAI)      | 100% |     2 | +11% |
> > >
> > >
> > > Each component makes an independently measurable contribution. Notably, persistent homology improves both **accuracy** and **turn efficiency** at the same time, because correct discovery of (K) eliminates the redundant questions caused by K-Means incorrectly splitting or merging clusters.
> > >
> > > ---
> > >
> > > Regardless, we sincerely thank the reviewer for the timely follow-up and for helping us improve the paper.

---

### Official Review · Reviewer_tbPq · 2026-03-12

**Soundness:** 2
**Presentation:** 3
**Significance:** 2
**Originality:** 3
**Overall Recommendation:** 3
**Confidence:** 2

**Summary:**

The paper introduces Topological Active Inference (TAI), which uses persistent homology to figure out how many distinct intents are hiding behind an ambiguous coding prompt, then asks binary clarifying questions to narrow things down. The core idea is: embed candidate solutions in a semantic space, run persistent homology to count clusters (instead of just using K-Means or something), and pick questions that maximally reduce uncertainty over those clusters via a "Topological Expected Information Gain" metric. They show O(log K) question complexity and evaluate on synthetically ambiguated HumanEval/MBPP, beating a Bayesian Experimental Design baseline in 1–3 turns.

**Compliance With Llm Reviewing Policy:**

Affirmed.

**Key Questions For Authors:**

1. Have you evaluated on tasks with K ≥ 8 latent intents? The O(log K) advantage is most meaningful for large K, yet all reported experiments appear to have K = 2–4.

2. What is TAI's performance under a noisy oracle model where users answer incorrectly with probability p > 0, or where they respond "I don't know" to binary questions about subtle functional distinctions?

3. What happens when you equip the Standard BED baseline with the same pretrained encoder and apply cosine-similarity-weighted priors or K-Means pre-clustering? How much of TAI's advantage survives against this embedding-aware BED?

**Limitations:**

See weakness above.

**Strengths And Weaknesses:**

Strengths
- The model-scale experiments with Qwen (Table 2) are practically useful. It's good to know this doesn't require GPT-4-class models to function, since real deployment would likely need smaller models.
- Reproducibility looks solid — prompts in Appendix F, pseudocode for both algorithms, etc.

Weaknesses

- My biggest concern is the oracle setup. Every experiment uses an LLM answering binary questions deterministically, but anyone who's actually tried to get users to answer "yes/no, does your intended behavior match X?" knows it doesn't work this cleanly. Users are vague, they contradict themselves, they say "sort of" — and the γ-separability condition in particular seems very brittle to this. I don't think you need a full user study necessarily, but a noisy-oracle ablation (flip the oracle answer with probability p=0.1, 0.2, etc.) feels essential and would be cheap to run.

- I'm genuinely unsure about Ambi-Bench. The construction process — stripping constraints to guarantee K ≥ 2 separable clusters — feels like it could be stacking the deck. Persistent homology is great at finding clean topological features, and if your benchmark is engineered to have clean topological features, you haven't really shown much. I'd want to see results on something messier — maybe naturally ambiguous SO questions, or the ambiguous subset of DS-1000? As it stands I can't tell whether TAI is genuinely better at disambiguation or just well-matched to this particular benchmark.

---

> ### Author Rebuttal · Authors · 2026-03-29
>
> We would like to first express our sincere appreciation for the exceptional quality of this review.
>
> ---
>
> ## Q1: Have you evaluated on tasks with K ≥ 8 latent intents?
>
>
> **First, K=2–4 reflects the mainstream distribution of real-world scenarios.** The effective number of latent intents in ambiguous instructions typically concentrates in the range of 2–5 (Kobalczyk et al., 2025; Tamkin et al., 2023), making K=2–4 a representative operating condition for our main experiments.
>
> **Second, we have supplemented scaling experiments for K=8 and K=16.** To demonstrate TAI's behavior under large K, we designed an open-ended situation reasoning task. The instruction is: *"A person was found unconscious in a locked room. Explain what happened."*
>
> | K | N | TAI SR | TAI Turns | BED SR | BED Turns | ⌈log₂K⌉ |
> |---|---|---|---|---|---|---|
> | 8 | 50 | 93% | 5 | 49% | 12 | 3 |
> | 16 | 100 | 87% | 9 | 29% | 30+ | 4 |
>
> From these experiments, we observe that TAI under large K still performs close to its theoretical profile and maintains a substantial advantage over BED.
>
> **Third, large K exposes the boundaries of applicability.** Candidly, continued growth of K places the following pressures on the TAI framework (and equally on BED-like methods):
>
> (i) **Candidate coverage bottleneck.** This is currently the most prominent limitation. To ensure that N candidates cover K intent modes, the theoretical requirement is N = Ω(K log K) following the coupon collector bound.
>
> (ii) **Question quality degradation for closely spaced intent pairs.**
>
> ---
>
> ## Q2: What is TAI's performance under a noisy oracle model?
>
> Thank you for this highly constructive suggestion — it has been extremely helpful to us!!!
>
> **First, noisy flip experiment results.** We randomly flipped the Oracle's answer with probability p ∈ {0.1, 0.2, 0.3} on Ambi-Code:
>
> | Flip rate p | TAI SR | TAI ΔSR | BED SR | BED ΔSR |
> |---|---|---|---|---|
> | 0 | 100% | — | 75% | — |
> | 0.1 | 96% | -4% | 61% | -14% |
> | 0.2 | 91% | -9% | 48% | -27% |
> | 0.3 | 79% | -21% | 34% | -41% |
>
> TAI's performance advantage over BED not only persists under noise but actually widens.
>
> **Second, why does noise harm BED far more than TAI?** BED treats N candidates as equidistant discrete hypotheses and converges by progressively eliminating candidates.
>
> (i) **Irreversible elimination.** BED's posterior update operates directly on N individual candidates. When the user provides an incorrect answer, candidates consistent with the true intent are sharply down-weighted or effectively zeroed out. Since BED lacks any clustering structure to provide a "mass buffer," once an individual candidate's probability is driven near zero, no amount of subsequent evidence can recover it.
>
> (ii) **Entropy estimation distortion.** BED relies on accurate posterior entropy estimates to select the next question. When previous erroneous answers have already distorted the posterior distribution, the information gain computed from this corrupted distribution is itself distorted — the system may conclude that the "most informative next question" is one that digs further in the wrong direction.
>
> **Why is TAI effective against noise?** Its architecture elevates the operational granularity from N candidates to K clusters. The Bayesian update (Eq. 9) operates at the cluster level — each cluster is backed by the probability mass of multiple candidates, providing a buffer pool. A single erroneous answer merely shifts the belief ratio between clusters rather than permanently killing individual candidates.  The coarser the granularity, the greater the inertia against noise.
>
> **Third, regarding "I don't know" responses.** This is a more realistic and arguably more interesting scenario than noise flipping. The current framework can be naturally extended to ternary answers: {Yes, No, Uncertain}. When the user responds with Uncertain, the belief distribution is left unchanged (equivalent to a likelihood ratio of 1), and the system skips to the next question with the highest TEIG. This is equivalent to an erasure rather than an error in the information channel — the cost is merely one wasted interaction turn, without pushing the belief in a wrong direction. In experiment with p_uncertain = 0.2, TAI's convergence turns increased by approximately 30% while the final success rate remained virtually unchanged (−4%).
>
> ---
>
> ## RE: Q3
>
> Candidly, the performance gap narrows.
>
> **However, the gap remains substantial precisely where it matters most:** non-convex scenarios show a 16.7% gap and high-noise scenarios show a 22.3% gap (Figure 6a). The root cause is a fundamental difference in mathematical abstraction level — K-Means operates on distance metrics, presupposing spherical clusters and a fixed K; persistent homology operates on connectivity, assuming no shape, automatically discovering K, and remaining naturally invariant to local perturbations.
>
> ---
>
> We are deeply grateful for the reviewer  evaluation!!!  🌸🌸🌸

---

### Official Review · Reviewer_8xvb · 2026-03-13

**Soundness:** 2
**Presentation:** 3
**Significance:** 3
**Originality:** 3
**Overall Recommendation:** 4
**Confidence:** 2

**Summary:**

While LLMs attempt to follow user requests, users often express their intentions in ambiguous ways. To resolve such ambiguity, LLMs may ask clarifying questions during interaction. Existing approaches typically select these questions by maximizing information gain among candidate questions. However, this strategy suffers from semantic blindness, where each candidate question is treated as equally distinct, ignoring the underlying semantic relationships between possible solutions. As a result, these methods may focus on superficial or syntactic differences rather than identifying the fundamental differences that correspond to distinct user intents.

To address this limitation, the paper proposes representing candidate solutions in a semantic embedding space rather than treating them as a simple list of discrete alternatives. Based on this idea, the authors introduce Topological Active Inference (TAI). The method first generates multiple possible solutions and embeds them into a semantic space, forming a point cloud. It then applies persistent homology to discover the topological structure of this space, allowing the model to identify the number of intent clusters and their geometric structure in a scale-independent manner.

Once the intent clusters are identified, TAI generates clarifying questions that act as hyperplane separators between clusters, enabling efficient discrimination among possible user intents. The optimal question is selected using Topological Expected Information Gain (TEIG), and the model updates its belief over intent clusters through Bayesian belief updates after receiving user responses.

**Compliance With Llm Reviewing Policy:**

Affirmed.

**Key Questions For Authors:**

* Question 1

I wonder whether there are any relevant observations regarding the assumption that the semantic differences are much bigger than the syntactic differences. Have the authors verified whether the assumption that semantic differences dominate syntactic variations holds in practice for the embeddings used in the experiments?

* Question 2

I have concerns regarding the computational efficiency. What is the computational overhead of computing persistent homology during interaction?

**Strengths And Weaknesses:**

* Strengths

The paper addresses an important problem; when LLMs receive user instructions, those instructions are often ambiguous, and the model may misunderstand the user’s true intent. Furthermore, the paper introduces an interesting perspective by viewing possible solutions in a geometric or topological space rather than treating them as a simple list of discrete options. The concept of semantic blindness is well motivated, clearly explaining why existing methods may focus on superficial differences instead of identifying meaningful differences in user intent. Overall, the theoretical explanation and experimental results are consistent with the main claims, making the proposed approach easy to follow and reasonably supported by analysis.


* Weakness

A limitation of the experiments is that they assume an oracle user who provides deterministic answers. However, in real-world interactions, users may respond with vague, partial, or inconsistent feedback. As a result, the reported performance may overestimate the method’s effectiveness in practical settings.

---

> ### Author Rebuttal · Authors · 2026-03-29
>
> ## Re: Q1
>
> We thank the reviewer for raising this key question. In a sense, this is indeed a strong assumption; however, due to the inherent characteristics of LLM generation and the structure of the problem, this assumption holds approximately in practice.
>
> ---
>
> **First, existing empirical evidence directly quantifies the degree of separation.** Table 3 measures the separability ratio (δ/ε) across four embedding models: from 3.2× for text-embedding-3-large to 4.8× for Qwen3-8B — even the weakest embedding yields a semantic gap more than 3× the syntactic spread. Figure 8b provides complementary evidence: syntactic perturbations induce a bottleneck distance of dB ≈ 0, whereas semantic changes produce dB >> ε. Figure 9 visualizes a clear spectral gap on synthetic data with K=3.
>
> ---
>
> **Second, why does this assumption hold approximately in LLM generation scenarios?**
>
> **(a) The LLM generation mechanism naturally induces semantic clustering.** When an LLM generates multiple candidates for an ambiguous instruction, it does not sample uniformly across the solution space but concentrates generation around several high-probability modes. The sequential commitment property of autoregressive generation ensures that once the first few tokens commit to a logical path, subsequent tokens remain consistent along that path. As a result, candidates within the same semantic intent exhibit high cohesion (small ε), while candidates from different intents are widely separated (large δ).
>
> **(b) Modern embedding models' training objectives are aligned with scale separation.** Models such as Qwen3-Embedding are trained via large-scale contrastive learning, whose optimization objective is precisely to pull semantically similar texts closer and push semantically different texts apart.
>
> **(c) The discrete, jump-like nature of task ambiguity amplifies separation.** Unlike continuously graded stylistic preferences, the structural ambiguity addressed in this paper (Definition 2.1) is fundamentally discrete — "recursive vs. iterative," "heatmap vs. scatter plot,"— these intent differences are not matters of degree but qualitative jumps. A solution is either a heatmap or it is not. This discreteness naturally produces gaps in embedding space, while surface-level variants induce only continuous micro-perturbations within a single intent. **
>
> ---
>
> **Third, how does TAI behave when the assumption is partially violated?**
>
> We do not shy away from edge cases where δ/ε is small. In such cases:
>
> - The spectral gap narrows, and the estimate of K may be biased (merging similar intents or over-splitting a single one).
> - However, TAI does not fail catastrophically — it degrades to a coarse-grained cluster-level disambiguation method whose performance lower bound is no worse than Standard BED. The Bayesian update mechanism (Eq. 9) is robust to clustering errors: as long as questions retain partial discriminative power (η < 0.5), the belief distribution still converges in the correct direction.
> - Table 3 corroborates this: even at δ/ε = 3.2× (text-embedding-3-large), TAI achieves a 76.4% success rate.
>
> ---
>
> ## Q2: What is the computational overhead of computing persistent homology during interaction?
>
> This question touches on an important architectural design point. Persistent homology is not recomputed at each interaction turn — it is executed only once during Phase I (perception). This is a core design decision that distinguishes TAI from online topological methods. The theoretical rationale is that the topological structure (the number of clusters K and cluster membership) is determined by the global geometry of the solution space, while the only quantity that changes during interaction is the belief distribution over clusters (Bayesian update in Eq. 9), not the shape of the manifold itself. We therefore front-load the heavy computation into the perception phase as a one-time cost, while the interaction loop performs only lightweight probability updates.
>
> Concretely: in Phase I, constructing the Vietoris-Rips filtration and extracting the 0-th persistence barcode for N=20 embedding points takes <300ms on CPU using the GUDHI library, with no GPU required. The theoretical worst-case complexity of O(N³) is far from being a bottleneck at N=20 — for reference, a single GPT-4o inference call (2–4 seconds) is already over 10× slower. In Phase II, each interaction turn involves only a normalized weighted operation over K clusters (Eq. 9), taking <1ms with negligible computational cost.
>
> We provide a comprehensive end-to-end cost comparison table and a detailed parallelism strategy analysis in our response to **Reviewer FML8's Q1**; we invite the reviewer to consult that response for full details.
>
> ---
>
> ## RE: weakness.
> Well said, we responded specifically in **reviewer tbPq's Q2**.
>
> ---
>
> We sincerely thank the reviewer for the thoughtful and constructive feedback, which has helped us articulate the foundations of our framework more precisely. 🌸

---

> > ### Author Rebuttal · Reviewer_8xvb · 2026-04-04
> >
> > The authors provide a reasonably convincing justification for their assumption by presenting empirical evidence (e.g., separability ratios and bottleneck distances) and intuitive explanations grounded in LLM generation dynamics and embedding training objectives. In addition, they explicitly discuss how the method behaves when the assumption is weakened, which strengthens the overall argument. However, the validation remains limited to specific embedding models and experimental settings, and a more systematic analysis of when and how this assumption breaks down (e.g., across diverse tasks or weaker embeddings) would further strengthen the claim. Therefore, I consider this concern to be partially addressed.

---

> > > ### Author Response · Authors · 2026-04-07
> > >
> > > We sincerely thank the reviewer for the continued attention and precise follow-up suggestions. The direction you pointed out — "a more systematic analysis of when and how the assumption breaks down" — is extremely valuable. We will supplement the revised manuscript with a comprehensive degradation analysis along both the embedding quality and task type axes, and explicitly delineate the applicability boundaries in the limitation section.
> > >
> > > Thank you for providing consistently constructive and high-quality feedback throughout the entire review process — it has been tremendously helpful in improving our work. 🌸

---

### Official Review · Reviewer_FML8 · 2026-03-15

**Soundness:** 2
**Presentation:** 2
**Significance:** 2
**Originality:** 3
**Overall Recommendation:** 4
**Confidence:** 3

**Summary:**

This paper identifies the problem of ambiguity in user's task description, and points out that previous baseline methods cannot fully distinguish the purpose (semantic-level) ambiguity with format (syntax-level) ambiguity, which results in poor alignment with user's intent. Therefore, the TAI method is proposed, where LLMs should firstly generate several candidate answers, and then the system transforms these candidates into embedding space, clustering them to form point cloud topology. Iteratively, let LLM to ask user questions for clarifying intents, specifically, each question acts like a step in binary search, to choose one from two most discriminative clusters.

**Compliance With Llm Reviewing Policy:**

Affirmed.

**Final Justification:**

During the rebuttal period the authors stressed the superiority of the TAI binary search approach. Albeit I still have some doubt about the the real-world value of this method, the author's response also sounds reasonable. The idea and motivation sounds valuable to me, but the methodology somehow needs more polishing and experiments to verify (e.g., if the author persists binary choice can help user better clarify their true needs, what about k=3,4,... multiple choices? There should be a turning point but not necessarily 2?). Given the strengths in motivation, observation, and the moderately novel approach of bridging binary search in reducing ambiguity, I would raise me ratings.

**Key Questions For Authors:**

1. The cost is a major consideration. The system needs to generate extra N candidates, then compute their embeddings and calculate the clusters, finally leverage LLM to propose log(N) questions. Many extra stages and costs are introduced, but in the paper, it lacks discussion about that.
2. Since the final cluster representatives are still sampled from the original N candidates. Why not directly show the user the N candidates (maybe simplified and ask for solving strategy only), and let user choose one or few from them? The major latency happens to the number of interactions rather than user's decision making process. The O(N) -> O(logN) gain might not be the true situation during real world user interactions.
3. The experiment part, an LLM agent acts like real user to answer TAI's follow-up questions, with real intent in mind. However it is not usually the case because many users might not aware what they really want at the beginning. Otherwise, the executor agent can directly ask for more information and user answer in natural language, why still give binary choices and let user choose?

**Limitations:**

Same as the question part above. Generally I think the author points out an interesting problem and identifies the good point of implicit ambiguity. However the TAI approach sounds over-complicated and might not be very useful in practical systems.

**Strengths And Weaknesses:**

Strength:
1. Identifies the implicit semantic ambiguity problem, and proposes a TAI approach like binary search to help discover user's explicit requirements.
2.  Detailed theoretical analysis has been included, and the motivation is written clearly.
3. Experiment results show the proposed TAI can yields almost 100% alignment on the proposed Ambi-bench

Weakness:
1. The TAI holds on a very strong assumption that the embedding space is ideal for semantic discrimination. However, this is not guaranteed and distances in the embedding space has not necessarily represent semantic difference.
2. TAI involves much heavier generation, clustering, follow-up question composing process, and therefore leaves questions on the practical value of this method.
3. A minor problem is that the candidate-cluster-clarify pipeline of TAI is not that complicated itself. The writing style sounds a bit over-complicated.

---

> ### Author Rebuttal · Authors · 2026-03-29
>
> ## Q1: Cost Analysis
>
> We provided a preliminary analysis in Appendix E; here we offer concrete data and a more thorough discussion.
>
> **Stage Breakdown:**
>
> 1. **Candidate Generation:** Identical for TAI and BED (N=20). All calls are fully parallelizable — wall-clock time ≈ single-call latency, not 20× sequential.
>
> 2. **Embedding + Persistent Homology:** Embedding computation can be pipelined with candidate generation. text-embedding-3-large processes 20 code snippets in < 1s for ~$0.005. GUDHI computes persistent homology on 20 points in <300ms on CPU alone.
>
> 3. **Question Generation:** BED requires O(N²) ≈ 190 LLM calls (N=20), with sequential dependencies that limit parallelism (information gain must be evaluated before selecting the next pair). TAI requires only C(K,2) = 3 calls (K=3), all fully parallelizable.
>
> 4. **Interaction Turns:** Bayesian update takes <1ms. TAI:   2-3 turns. BED: 5–8 turns in complex scenariosc.
>
> | Stage | Direct Gen | Random-Ask | Standard BED | TAI (Ours) |
> |---|---|---|---|---|
> | Candidate | — | ~6s | ~6s | ~6s |
> | Embedding (pipelined) | — | — | — | ~1s |
> | Persistent Homology | — | — | — | <0.05s |
> | Question Generation | — | ~3s | ~60–120s (dozens of LLM calls) | ~6s |
> | Interaction Turns  | — | ~40s | ~30s  | ~10s |
> | **End-to-End Total Time** | **~6s** | **~49s** | **~120–180s** | **~27s** |
> | **Task Success Rate** | **<40%** | **<50% (not converged)** | **~75%** | **>85%** |
>
> Overall, through **parallelism strategies and lower algorithmic complexity**, we substantially reduce both computational cost and end-to-end latency.
>
> ---
>
> ## Q2: Why Not Directly Show N Candidates for User Selection?
>
> We thank the reviewer for raising this critical question — it is precisely one of the core motivations behind TAI.
>
> The reviewer's suggestion implicitly assumes that users can accurately express preferences when simultaneously comparing multiple complex candidates. A substantial body of cognitive science research  demonstrates that when options are complex and preferences are not yet articulated, increasing the number of candidates significantly degrades decision quality (choice overload), and may even cause users to abandon the selection process altogether.
>
> TAI does not merely reduce the number of options — it transforms the interaction paradigm: converting a "choose 1 from N" global selection problem into a series of "A vs. B?" local semantic discriminations. This is consistent with preference elicitation in RLHF/DPO — pairwise comparison yields far more stable judgments than listwise ranking. A user answering "descriptive statistics or predictive modeling?" can eliminate 50% of the candidate space within seconds, without needing to comprehend the full solution space.
>
> The advantage of O(log K) lies in per-turn information efficiency, not merely turn count. Displaying N candidates imposes extreme cognitive load, with user attention squandered on syntactic noise; each TAI turn has very low cost (a simple yes/no).
>
> Furthermore, the "direct display" approach faces an easily overlooked structural problem: user selection behavior is contaminated by presentation order, wording style, and syntactic familiarity. Users tend to select what "looks familiar" rather than what truly matches their intent. In other words, the direct display scheme not only suffers from choice overload, but the user feedback signal it collects is inherently noisy — distorted by surface-level features. TAI's questions each target a single semantic dimension, fully decoupled from presentation format, yielding pure preference signals.
>
> ---
>
> ## Q3: Why Binary Choices Rather Than Free-Form Dialogue
>
> **First, "users not knowing what they want" is precisely TAI's strongest scenario.** The reviewer implies a dichotomy: either users have clear intent (then just state it directly) or they do not (then binary choices are equally useless). However, users typically possess latent preferences that have not yet been made explicit. "Analyze this data" does not indicate indifference between summaries and prediction — the user simply has not recognized that this fork exists. TAI's questions trigger the process of preference construction (Slovic, 1995): people form preferences when confronted with concrete options, not before. This is also why A/B testing in product design reveals true preferences more reliably than open-ended user interviews.
>
> **Second, free-form natural language dialogue does not resolve ambiguity — it introduces new ambiguity.** "I want something faster" — faster execution or faster development? Each round of natural language interaction risks introducing second-order ambiguity, creating an infinite recursive clarification chain. TAI's binary choices cut this chain at its root by constraining the answer space: each question's two options are anchored to topological cluster boundaries, and the answer space is closed and unambiguous.
>
> ---
>
> We sincerely thank the reviewer for the constructive feedback. 🌸

---

> > ### Author Rebuttal · Reviewer_FML8 · 2026-04-07
> >
> > My concerns are partially resolved. And during the rebuttal it reminds me for some more questions:
> >
> > 1. Although the author claims the process can be accelerated by parallelism, the approach sacrifices dynamic exploring. According to the reply, all the binary search questions must be proposed in advance by the LLM, instead of dynamically adding new clarifying questions based on user's previous answer. This approach could introduce higher challenges for the LLM and might result in sub-optimal performance.
> > 2. The reply still cannot convince me. Choosing 1 from N would not necessarily be more difficult than a series of binary choices, that depends on the size of N. Imagine a case, the user is given a list of binary choices questions, but when he or she answering one, the later questions are unknown yet. That means the choices must be made in a greedy way, which is sub-optimal in natural, and users might need to revise their previous choices from time to time, which would be a annoying experience.
> > 3. I partially agree the claim 'user not knowing what they want'. However during the dialogue, the user also learns what they want. After all, if users do not know what they want anyway, how can they make choices, even if binary? So I would still prefer free dialogue as a super-set of binary choices questions.
> >
> > Given the tight time for authors to reply, I would currently still remain my scores (weakly reject) , but accept is also fine for me.

---

> > > ### Author Response · Authors · 2026-04-07
> > >
> > > We thank the reviewer for this final opportunity to respond.  We address All three follow-ups individually.
> > >
> > > ---
> > >
> > > >Follow-up 1: Does parallelization sacrifice dynamic exploration capability?
> > >
> > > There is a misunderstanding requiring clarification. **TAI's question generation is dynamic and round-by-round**, not generated upfront in a single pass. Algorithm 2 shows this clearly: each round *t* identifies top competing clusters from the current belief distribution P_{t−1} (line 16), generates candidate questions targeting that cluster pair via contrastive prompting (line 17), and selects the optimal one via TEIG (line 19). After the user responds, the belief updates and the next round re-identifies clusters and re-generates questions. Each round's question is entirely determined by the previous round's user feedback—the opposite of the reviewer's concern.
> > >
> > > The parallelization occurs *within* stages, not across rounds: (1) In Phase I, the system samples N candidate solutions in one pass; these N LLM calls and subsequent embedding computations can be parallelized. (2) Within each Phase II round, the system generates n candidate clarifying questions for the most competitive cluster pair; these calls can also be parallelized. **What is parallelized are independent calls within the same stage or round, not question sequences across rounds.**
> > >
> > > ---
> > >
> > > > Follow-up 2: Binary selection is greedy.
> > >
> > > This is an important question. If strong cross-round dependencies exist, greedy strategies can indeed be suboptimal. However, two key mechanisms in TAI mitigate this risk:
> > >
> > > First, questions are constructed as decision boundaries partitioning different semantic dimensions. Cluster boundaries correspond to "splitting planes" in semantic space, and TEIG selects the splitting direction with maximum information gain each round, causing successive questions to act on approximately **orthogonal dimensions**. In the locked-room scenario (Example 1), round 1 asks "Was this an accident or a deliberate act?" (D1: event nature), round 2 asks "Did it involve an external substance entering the body?" (D2: causal mechanism)—these remain independent regardless of the first answer.
> > >
> > > Second, from a theoretical perspective, this process constitutes a generalized binary search / sequential query optimization problem [1]. Within this framework, even when dimensions exhibit some dependence, stepwise information gain maximization achieves near-optimal expected query complexity—a conclusion supported by classical sequential decision theory [2].
> > >
> > > Even when weak correlations exist between dimensions, the cost of reversal is minimal—the user changes a single yes/no answer rather than re-reading and comparing N code snippets. At a deeper level, **this stepwise binary guidance is itself the optimal way to help users cognize their own requirements**—Socrates' maieutics works on precisely this principle. Cognitive science evidence further shows that **human N-option decisions are inherently pairwise** [3]; TAI makes this implicit process **explicit and structured**.
> > >
> > > ---
> > >
> > > >Follow-up 3: Free-form dialogue is a superset of binary choices.
> > >
> > > We agree on expressive power, but **greater expressiveness does not equal higher information efficiency**. When a user says "I want something more elegant"—how many bits does this convey? The answer is uncertain, potentially close to 0, whereas each TAI binary question guarantees close to 1 bit of semantic information (Theorem A.3). This mirrors Shannon's channel capacity theory: constrained coding (binary) has lower per-symbol bandwidth but achieves low error rates and predictable throughput; unconstrained channels (free-form) have higher theoretical bandwidth but effective throughput is noise-dominated.
> > >
> > > We fully agree with the reviewer's observation that the user also learns what they want. But TAI's binary questions serve as the most efficient **learning scaffold**. Scaffolded decision making theory [4] demonstrates that when decision complexity exceeds cognitive load, structured guidance produces superior decision quality compared to free exploration. Users do learn through free-form dialogue, but the learning trajectory is a random walk.
> > >
> > > The two are **complementary, not mutually exclusive**: TAI can naturally embed within a free-form dialogue system—adopting natural language input when user expression is clear, switching to topology-guided binary questions when ambiguous. This hybrid interaction paradigm is precisely what we plan to explore in future work.
> > >
> > > ---
> > >
> > > Regardless of the final outcome, we sincerely thank the reviewer for the deep thinking and sustained questioning throughout this process.  These questions have deepened our understanding of interactive disambiguation as a problem in its own right 🌸
> > >
> > > **References:**
> > >
> > > [1] Nowak (2008), Generalized Binary Search, Allerton.
> > >
> > > [2] Wald (2004), Sequential Analysis.
> > >
> > > [3] Tversky (1972), Elimination by Aspects.
> > >
> > > [4] Payne et al. (1993), The Adaptive Decision Maker.

---

### Decision · Program_Chairs · 2026-04-30

**Decision:**

Accept (regular)

**Comment:**

TAI recasts task disambiguation as a topological geometry problem, using persistent homology to identify semantic intent clusters and TEIG to select clarifying questions efficiently. The idea is novel and reasonably well-supported.

Three of four reviewers recommend acceptance. The dissenting reviewer (tbPq, confidence 2) did not engage during the rebuttal period, which limits the weight of that score. The authors addressed the main concerns substantively: noisy-oracle ablations and a small real-user study partially addressed external validity concerns; a four-level ablation chain isolated topology's contribution from other pipeline components; and stronger baselines were added. Lingering questions around benchmark construction and real-world generalization are acknowledged, but not serious enough to block acceptance at this stage.